# ANALYSIS OF QUANTIZED MODELS

**Lu Hou**[1]**, Ruiliang Zhang**[1,2]**, James T. Kwok**[1]
[1]Department of Computer Science and Engineering
Hong Kong University of Science and Technology
Hong Kong
{lhouab,jamesk}@cse.ust.hk
[2]TuSimple
ruiliang.zhang@tusimple.ai

## ABSTRACT

Deep neural networks are usually huge, which significantly limits the deployment on low-end devices. In recent years, many weight-quantized models have been proposed. They have small storage and fast inference, but training can still be time-consuming. This can be improved with distributed learning. To reduce the high communication cost due to worker-server synchronization, recently gradient quantization has also been proposed to train deep networks with full-precision weights. In this paper, we theoretically study how the combination of both weight and gradient quantization affects convergence. We show that (i) weight-quantized models converge to an error related to the weight quantization resolution and weight dimension; (ii) quantizing gradients slows convergence by a factor related to the gradient quantization resolution and dimension; and (iii) clipping the gradient before quantization renders this factor dimension-free, thus allowing the use of fewer bits for gradient quantization. Empirical experiments confirm the theoretical convergence results, and demonstrate that quantized networks can speed up training and have comparable performance as full-precision networks.

## 1 INTRODUCTION

Deep neural networks are usually huge. The high demand in time and space can significantly limit deployment on low-end devices. To alleviate this problem, many approaches have been recently proposed to compress deep networks. One direction is network quantization, which represents each network weight with a small number of bits. Besides significantly reducing the model size, it also accelerates network training and inference. Many weight quantization methods aim at approximating the full-precision weights in each iteration (Courbariaux et al., 2015; Lin et al., 2016; Rastegari et al., 2016; Li & Liu, 2016; Lin et al., 2017; Guo et al., 2017). Recently, loss-aware quantization minimizes the loss directly w.r.t. the quantized weights (Hou et al., 2017; Hou & Kwok, 2018; Leng et al., 2018), and often achieves better performance than approximation-based methods.

Distributed learning can further speed up training of weight-quantized networks (Dean et al., 2012). A key challenge is on reducing the expensive communication cost incurred during synchronization of the gradients and model parameters (Li et al., 2014a;b). Recently, algorithms that sparsify (Aji & Heafield, 2017; Wangni et al., 2017) or quantize the gradients (Seide et al., 2014; Wen et al., 2017; Alistarh et al., 2017; Bernstein et al., 2018) have been proposed.

In this paper, we consider quantization of both the weights and gradients in a distributed environment. Quantizing both weights and gradients has been explored in the DoReFa-Net (Zhou et al., 2016), QNN (Hubara et al., 2017), WAGE (Wu et al., 2018) and ZipML (Zhang et al., 2017). We differ from them in two aspects. First, existing methods mainly consider learning on a single machine, and gradient quantization is used to reduce the computations in backpropagation. On the other hand, we consider a distributed environment, and use gradient quantization to reduce communication cost and accelerate distributed learning of weight-quantized networks. Second, while DoReFa-Net, QNN and WAGE show impressive empirical results on the quantized network, theoretical guarantees are not provided. ZipML provides convergence analysis, but is limited to stochastic weight quantization, square loss with the linear model, and requires the stochastic gradients to be unbiased. This can

be restrictive as most state-of-the-art weight quantization methods (Rastegari et al., 2016; Lin et al., 2016; Li & Liu, 2016; Guo et al., 2017; Hou et al., 2017; Hou & Kwok, 2018) are deterministic, and the resultant stochastic gradients are biased.

In this paper, we relax the restrictions on the loss function, and study in an online learning setting how the gradient precision affects convergence of weight-quantized networks in a distributed environment. The main findings are:

1. With either full-precision or quantized gradients, the average regret of loss-aware weight quantization does not converge to zero, but to an error related to the weight quantization resolution $\Delta_w$ and dimension $d$. The smaller the $\Delta_w$ or $d$, the smaller is the error (Theorems 1 and 2).
2. With either full-precision or quantized gradients, the average regret converges with a $O(1/\sqrt{T})$ rate to the error, where $T$ is the number of iterations. However, gradient quantization slows convergence (relative to using full-precision gradients) by a factor related to gradient quantization resolution $\Delta_g$ and $d$. The larger the $\Delta_g$ or $d$, the slower is the convergence (Theorems 1 and 2). This can be problematic when (i) the weight quantized model has a large $d$ (e.g., deep networks); and (ii) the communication cost is a bottleneck in the distributed setting, which favors a small number of bits for the gradients, and thus a large $\Delta_g$.
3. For gradients following the normal distribution, gradient clipping renders the speed degradation mentioned above dimension-free. However, an additional error is incurred. The convergence speedup and error are related to how aggressive clipping is performed. More aggressive clipping results in faster convergence, but a larger error (Theorem 3).
4. Empirical results show that quantizing gradients significantly reduce communication cost, and gradient clipping makes speed degradation caused by gradient quantization negligible. With quantized clipped gradients, distributed training of weight-quantized networks is much faster, while comparable accuracy with the use of full-precision gradients is maintained (Section 4).

**Notations.** For a vector $\mathbf{x}$, $\sqrt{\mathbf{x}}$ is the element-wise square root, $\mathbf{x}^2$ is the element-wise square, $\text{Diag}(\mathbf{x})$ returns a diagonal matrix with $\mathbf{x}$ on the diagonal, and $\mathbf{x} \odot \mathbf{y}$ is the element-wise multiplication of vectors $\mathbf{x}$ and $\mathbf{y}$. For a matrix $\mathbf{Q}$, $\|\mathbf{x}\|_{\mathbf{Q}}^2 = \mathbf{x}^\top \mathbf{Q} \mathbf{x}$. For a matrix $\mathbf{X}$, $\sqrt{\mathbf{X}}$ is the element-wise square root, and $\text{diag}(\mathbf{X})$ returns a vector extracted from the diagonal elements of $\mathbf{X}$.

## 2 PRELIMINARIES

### 2.1 ONLINE LEARNING

Online learning continually adapts the model with a sequence of observations. It has been commonly used in the analysis of deep learning optimizers (Duchi et al., 2011; Kingma & Ba, 2015; Reddi et al., 2018). At time $t$, the algorithm picks a model with parameter $\mathbf{w}_t \in \mathcal{S}$, where $\mathcal{S}$ is a convex compact set. The algorithm then incurs a loss $f_t(\mathbf{w}_t)$. After $T$ rounds, the performance is usually evaluated by the regret $R(T) = \sum_{t=1}^T f_t(\mathbf{w}_t) - f_t(\mathbf{w}^*)$ and average regret $R(T)/T$, where $\mathbf{w}^* = \arg\min_{\mathbf{w} \in \mathcal{S}} \sum_{t=1}^T f_t(\mathbf{w})$ is the best model parameter in hindsight.

### 2.2 WEIGHT QUANTIZATION

In BinaryConnect (Courbariaux et al., 2015), each weight is binarized using the sign function either deterministically or stochastically. In ternary-connect (Lin et al., 2016), each weight is stochastically quantized to $\{-1, 0, 1\}$. Stochastic weight quantization often suffers severe accuracy degradation, while deterministic weight quantization (as in the binary-weight-network (BWN) (Rastegari et al., 2016) and ternary weight network (TWN) (Li & Liu, 2016)) achieves much better performance.

In this paper, we will focus on loss-aware weight quantization, which further improves performance by considering the effect of weight quantization on the loss. Examples include loss-aware binarization (LAB) (Hou et al., 2017) and loss-aware quantization (LAQ) (Hou & Kwok, 2018). Let the full-precision weights from all $L$ layers in the deep network be $\mathbf{w}$. The corresponding quantized weight is denoted $Q_w(\mathbf{w}) = \hat{\mathbf{w}}$, where $Q_w(\cdot)$ is the weight quantization function. At the $(t+1)$th iteration, the second-order Taylor expansion of $f_t(\hat{\mathbf{w}})$, i.e., $f_t(\hat{\mathbf{w}}_t) + \nabla f_t(\hat{\mathbf{w}}_t)^\top (\hat{\mathbf{w}} - \hat{\mathbf{w}}_t) + \frac{1}{2}(\hat{\mathbf{w}} - \hat{\mathbf{w}}_t)^\top \mathbf{H}_t (\hat{\mathbf{w}} - \hat{\mathbf{w}}_t)$ is minimized w.r.t. $\hat{\mathbf{w}}$, where $\mathbf{H}_t$ is the Hessian at $\hat{\mathbf{w}}_t$. A direct computation of

$\mathbf{H}_t$ is expensive. In practice, this is approximated by $\text{Diag}(\sqrt{\hat{\mathbf{v}}_t})$, where $\hat{\mathbf{v}}_t$ is the moving average:

$$\hat{\mathbf{v}}_t = \beta \hat{\mathbf{v}}_{t-1} + (1 - \beta)\hat{\mathbf{g}}_t^2 = \sum_{j=1}^{t} (1 - \beta)\beta^{t-j}\hat{\mathbf{g}}_j^2, \tag{1}$$

with $\mathbf{g}_t$ the stochastic gradient, $\beta \simeq 1$, and is readily available in popular deep network optimizers such as RMSProp and Adam. $\text{Diag}(\sqrt{\hat{\mathbf{v}}_t})$ is also an estimate of $\text{Diag}(\sqrt{\text{diag}(\mathbf{H}_t^2)})$ (Dauphin et al., 2015). Computationally, the quantized weight is obtained by first performing a preconditioned gradient descent $\mathbf{w}_{t+1} = \mathbf{w}_t - \eta_t \text{Diag}(\sqrt{\hat{\mathbf{v}}_t})^{-1}\hat{\mathbf{g}}_t$, followed by quantization via solving the following problem:

$$\hat{\mathbf{w}}_{t+1} = Q_w(\mathbf{w}_{t+1}) = \arg\min_{\hat{\mathbf{w}}} \|\mathbf{w}_{t+1} - \hat{\mathbf{w}}\|^2_{\text{Diag}(\sqrt{\hat{\mathbf{v}}_t})} \quad \text{s.t.} \quad \hat{\mathbf{w}} = \alpha\mathbf{b}, \ \alpha > 0, \ \mathbf{b} \in (\mathcal{S}_w)^d. \tag{2}$$

For simplicity of notations, we assume that the same scaling parameter $\alpha$ is used for all layers. Extension to layer-wise scaling is straightforward. For binarization, $\mathcal{S}_w = \{-1, +1\}$, the weight quantization resolution is $\Delta_w = 1$, and a simple closed-form solution is obtained in (Hou et al., 2017). For $m$-bit linear quantization, $\mathcal{S}_w = \{-M_k, \ldots, -M_1, M_0, M_1, \ldots, M_k\}$, where $k = 2^{m-1} - 1$, $0 = M_0 < \cdots < M_k$ are uniformly spaced, with weight quantization resolution $\Delta_w = M_{r+1} - M_r$. An efficient approximate solution of (2) is obtained in (Hou & Kwok, 2018).

## 2.3 GRADIENT QUANTIZATION

In a distributed learning environment with data parallelism, the main bottleneck is often on the communication cost due to gradient synchronization. By quantizing the gradients before synchronization (Seide et al., 2014; Wen et al., 2017; Alistarh et al., 2017), this cost can be significantly reduced. For example, assuming that the full-precision gradient is 32-bit, the communication cost can be reduced $32/m$ times when gradients are quantized to $m$ bits.

Most recent gradient quantization methods (Wen et al., 2017; Alistarh et al., 2017; Zhang et al., 2017) require the quantized gradient to be unbiased, and thus use stochastically quantized gradients. On the other hand, deterministic gradient quantization makes the quantized gradient biased, and the resultant analysis more complex. In this paper, we consider the more general $m$-bit stochastic linear quantization (Alistarh et al., 2017):

$$Q_g(\mathbf{g}_t) = s_t \cdot \text{sign}(\mathbf{g}_t) \odot \mathbf{q}_t, \tag{3}$$

where $s_t = \|\mathbf{g}_t\|_\infty$, $\mathbf{q}_t \in (\mathcal{S}_g)^d$, and $\mathcal{S}_g = \{-B_k, \ldots, -B_1, B_0, B_1, \ldots, B_k\}$, with $k = 2^{m-1} - 1$, $0 = B_0 < B_1 < \cdots < B_k$ are uniformly spaced. The gradient quantization resolution is defined as $\Delta_g = B_{r+1} - B_r$. The $i$th element $q_{t,i}$ in $\mathbf{q}_t$ is equal to $B_{r+1}$ with probability $(|g_{t,i}|/s_t - B_r)/(B_{r+1} - B_r)$, and $B_r$ otherwise. Here, $r$ is an index satisfying $B_r \le |g_{t,i}|/s_t < B_{r+1}$. Note that $Q_g(\mathbf{g}_t)$ is an unbiased estimator of $\mathbf{g}_t$.

# 3 PROPERTIES OF A QUANTIZED MODEL

In this section, we consider quantization of both weights and gradients in a distributed environment with $N$ workers using data parallelism. For easy illustration, we use the parameter server model (Li et al., 2014b) in Figure 1, though it also holds for other configurations such as the AllReduce model (Rabenseifner, 2004). At the $t$th iteration, worker $n \in \{1, 2, \ldots, N\}$ computes the full-precision gradient $\hat{\mathbf{g}}_t^{(n)}$ w.r.t. the quantized weight and quantizes $\hat{\mathbf{g}}_t^{(n)}$ to $\tilde{\mathbf{g}}_t^{(n)} = Q_g(\hat{\mathbf{g}}_t^{(n)})$. The quantized gradients are then synchronized and averaged at the parameter server as: $\tilde{\mathbf{g}}_t = \frac{1}{N}\sum_{n=1}^{N} \tilde{\mathbf{g}}_t^{(n)}$. The server updates the second moment $\tilde{\mathbf{v}}_t$ based on $\tilde{\mathbf{g}}_t$, and also the full-precision weight as $\mathbf{w}_{t+1} = \mathbf{w}_t - \eta_t \text{Diag}(\sqrt{\tilde{\mathbf{v}}_t})^{-1}\tilde{\mathbf{g}}_t$. The weight is quantized using loss-aware weight quantization to produce $\hat{\mathbf{w}}_{t+1} = Q_w(\mathbf{w}_{t+1})$, which is then sent back to all the workers.

## 3.1 ASSUMPTIONS

Analysis on quantized deep networks has only been performed on models with (i) full-precision gradients and weights quantized by stochastic weight quantization (Li et al., 2017; De Sa et al., 2018), or simple deterministic weight quantization using the sign (Li et al., 2017); (ii) full-precision weights and quantized gradients (Alistarh et al., 2017; Wen et al., 2017; Bernstein et al., 2018);

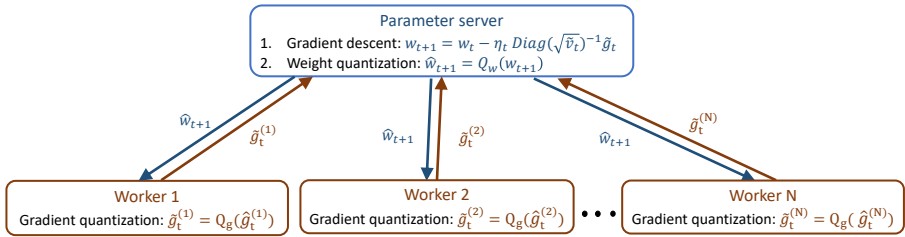

Figure 1: Distributed weight and gradient quantization with data parallelism.

(iii) quantized weights and quantized gradients (Zhang et al., 2017), but limited to stochastic weight quantization, square loss on linear model (i.e., $f_t(\mathbf{w}_t) = (\mathbf{x}_t^\top \mathbf{w}_t - y_t)^2$) in Section 2.1), and unbiased gradient.

In this paper, we study the more advanced loss-aware weight quantization, with both full-precision and quantized gradients. As it is deterministic and has biased gradients, the above analysis do not apply here. Moreover, we do not assume a linear model, and relax the assumptions on $f_t$ as:

(**A1**) $f_t$ is convex;

(**A2**) $f_t$ is twice differentiable with Lipschitz-continuous gradient; and

(**A3**) $f_t$ has bounded gradient, i.e., $\|\nabla f_t(\mathbf{w})\| \le G$ and $\|\nabla f_t(\mathbf{w})\|_\infty \le G_\infty$ for all $\mathbf{w} \in \mathcal{S}$.

These assumptions have been commonly used in convex online learning (Hazan, 2016; Duchi et al., 2011; Kingma & Ba, 2015) and quantized networks (Alistarh et al., 2017; Li et al., 2017). Obviously, the convexity assumption **A1** does not hold for deep networks. However, this facilitates analysis of deep learning models, and has been used in (Kingma & Ba, 2015; Reddi et al., 2018; Li et al., 2017; De Sa et al., 2018). Moreover, as will be seen, it helps to explain the empirical behavior in Section 4.

As in (Duchi et al., 2011; Kingma & Ba, 2015; Li et al., 2017), we assume that $\|\mathbf{w}_m - \mathbf{w}_n\| \le D$ and $\|\mathbf{w}_m - \mathbf{w}_n\|_\infty \le D_\infty$ for all $\mathbf{w}_m, \mathbf{w}_n \in \mathcal{S}$. Moreover, the learning rate $\eta_t$ decays as $\eta/\sqrt{t}$, where $\eta$ is a constant (Hazan, 2016; Duchi et al., 2011; Kingma & Ba, 2015; Li et al., 2017).

For simplicity of notations, we denote the full-precision gradient $\nabla f_t(\mathbf{w}_t)$ w.r.t. the full-precision weight by $\mathbf{g}_t$, and the full-precision gradient $\nabla f_t(Q_w(\mathbf{w}_t))$ w.r.t. the quantized weight by $\hat{\mathbf{g}}_t$. As $f_t$ is twice differentiable (Assumption **A2**), using the mean value theorem, there exists $p \in (0,1)$ such that $\mathbf{g}_t - \hat{\mathbf{g}}_t = \nabla f_t(\mathbf{w}_t) - \nabla f_t(\hat{\mathbf{w}}_t) = \nabla^2 f_t(\hat{\mathbf{w}}_t + p(\mathbf{w}_t - \hat{\mathbf{w}}_t))(\mathbf{w}_t - \hat{\mathbf{w}}_t)$. Let $\mathbf{H}'_t = \nabla^2 f_t(\hat{\mathbf{w}}_t + p(\mathbf{w}_t - \hat{\mathbf{w}}_t))$ be the Hessian at $\hat{\mathbf{w}}_t + p(\mathbf{w}_t - \hat{\mathbf{w}}_t)$. Moreover, let $\alpha = \max\{\alpha_1, \ldots, \alpha_T\}$, where $\alpha_t$ is the scaling parameter in (2) at the $t$th iteration.

### 3.2 WEIGHT QUANTIZATION WITH FULL-PRECISION GRADIENT

When only weights are quantized, the update for loss-aware weight quantization is

$$\mathbf{w}_{t+1} = \mathbf{w}_t - \eta_t \text{Diag}(\sqrt{\hat{\mathbf{v}}_t})^{-1} \hat{\mathbf{g}}_t,$$

where $\hat{\mathbf{v}}_t$ is the moving average of the (squared) gradients $\hat{\mathbf{g}}_t^2$ in (1).

**Theorem 1.** *For loss-aware weight quantization with full-precision gradients and $\eta_t = \eta/\sqrt{t}$,*

$$
\begin{aligned}
R(T) &\le \frac{D_\infty^2 \sqrt{dT}}{2\eta} \sqrt{\sum_{t=1}^{T} (1-\beta)\beta^{T-t} \|\hat{\mathbf{g}}_t\|^2} + \frac{\eta G_\infty \sqrt{d}}{\sqrt{1-\beta}} \sqrt{\sum_{t=1}^{T} \|\hat{\mathbf{g}}_t\|^2} \\
&\quad + \sqrt{L} D \sum_{t=1}^{T} \sqrt{\|\mathbf{w}_t - \hat{\mathbf{w}}_t\|_{\mathbf{H}'_t}^2},
\end{aligned}
\tag{4}
$$

$$
\frac{R(T)}{T} \le O\left(\frac{d}{\sqrt{T}}\right) + LD\sqrt{D^2 + \frac{d\alpha^2 \Delta_w^2}{4}}.
\tag{5}
$$

For standard online gradient descent with the same learning rate scheme, $R(T)/T$ converges to zero at the rate of $O(1/\sqrt{T})$ (Hazan, 2016). From Theorem 1, the average regret converges at the same rate, but only to a nonzero error $LD\sqrt{D^2 + \frac{d\alpha^2 \Delta_w^2}{4}}$ related to the weight quantization resolution $\Delta_w$ and dimension $d$.

## 3.3 Weight Quantization With Quantized Gradient

When both weights and gradients are quantized, the update for loss-aware weight quantization is

$$\mathbf{w}_{t+1} = \mathbf{w}_t - \eta_t \text{Diag}(\sqrt{\tilde{\mathbf{v}}_t})^{-1}\tilde{\mathbf{g}}_t,$$

where $\tilde{\mathbf{g}}_t$ is the stochastically quantized gradient $Q_g(\nabla f_t(Q_w(\mathbf{w}_t)))$. The second moment $\tilde{\mathbf{v}}_t$ is the moving average of the (squared) quantized gradients $\tilde{\mathbf{g}}_t^2$. The following Proposition shows that gradient quantization significantly blows up the norm of the quantized gradient relative to its full-precision counterparts. Moreover, the difference increases with the gradient quantization resolution $\Delta_g$ and dimension $d$.

**Proposition 1.** $\mathbf{E}(\|\tilde{\mathbf{g}}_t\|^2) \leq (\frac{1+\sqrt{2d-1}}{2}\Delta_g + 1)\|\hat{\mathbf{g}}_t\|^2$.

**Theorem 2.** *For loss-aware weight quantization with quantized gradients and $\eta_t = \eta/\sqrt{t}$,*

$$
\begin{aligned}
\mathbf{E}(R(T)) &\leq \frac{D_\infty^2\sqrt{dT}}{2\eta}\sqrt{\sum_{t=1}^{T}(1-\beta)\beta^{T-t}\mathbf{E}(\|\tilde{\mathbf{g}}_t\|^2)} + \frac{\eta G_\infty\sqrt{d}}{\sqrt{1-\beta}}\sqrt{\sum_{t=1}^{T}\mathbf{E}(\|\tilde{\mathbf{g}}_t\|^2)} \\
&\quad + \sqrt{L}D\sum_{t=1}^{T}\mathbf{E}(\sqrt{\|\mathbf{w}_t - \hat{\mathbf{w}}_t\|_{\mathbf{H}'_t}^2}), \qquad\qquad (6)
\end{aligned}
$$

$$
\mathbf{E}\left(\frac{R(T)}{T}\right) \leq O\left(\sqrt{\frac{1+\sqrt{2d-1}}{2}\Delta_g + 1}\cdot\frac{d}{\sqrt{T}}\right) + LD\sqrt{D^2 + \frac{d\alpha^2\Delta_w^2}{4}}. \qquad (7)
$$

The regrets in (4) and (6) are of the same form and differ only in the gradient used. Similarly, for the average regrets in (5) and (7), quantizing gradients slows convergence by a factor of $\sqrt{\frac{1+\sqrt{2d-1}}{2}\Delta_g + 1}$, which is a direct consequence of the blowup in Proposition 1. These observations can be problematic as (i) deep networks typically have a large $d$; and (ii) distributed learning prefers using a small number of bits for the gradients, and thus a large $\Delta_g$.

## 3.4 Weight Quantization with Quantized Clipped Gradients

To reduce convergence speed degradation caused by gradient quantization, gradient clipping has been proposed as an empirical solution (Wen et al., 2017). The gradient $\hat{\mathbf{g}}_t$ is clipped to $\text{Clip}(\hat{\mathbf{g}}_t)$, where

$$
\text{Clip}(\hat{g}_{t,i}) = \begin{cases} \hat{g}_{t,i} & |\hat{g}_{t,i}| \leq c\sigma, \\ \text{sign}(\hat{g}_{t,i})\cdot c\sigma & \text{otherwise}. \end{cases}
$$

Here, $c$ is a constant clipping factor, and $\sigma$ is the standard deviation of elements in $\hat{\mathbf{g}}_t$. The update then becomes

$$\mathbf{w}_{t+1} = \mathbf{w}_t - \eta_t \text{Diag}(\sqrt{\check{\mathbf{v}}_t})^{-1}\check{\mathbf{g}}_t,$$

where $\check{\mathbf{g}}_t \equiv Q_g(\text{Clip}(\hat{\mathbf{g}}_t)) \equiv Q_g(\text{Clip}(\nabla f_t(Q_w(\mathbf{w}_t))))$ is the quantized clipped gradient. The second moment $\check{\mathbf{v}}_t$ is computed using the (squared) quantized clipped gradient $\check{\mathbf{g}}_t^2$.

As shown in Figure 2(a) of (Wen et al., 2017), the distribution of gradients before quantization is close to the normal distribution. Recall from Section 3.3 that the difference between $\mathbf{E}(\|\tilde{\mathbf{g}}_t\|^2)$ of the quantized gradient $\tilde{\mathbf{g}}_t$ and the full-precision gradient $\|\hat{\mathbf{g}}_t\|^2$ is related to the dimension $d$. The following Proposition shows that $\mathbf{E}(\|\check{\mathbf{g}}_t\|^2)/\mathbf{E}(\|\hat{\mathbf{g}}_t\|^2)$ becomes independent of $d$ if $\hat{\mathbf{g}}_t$ follows the normal distribution and clipping is used.

**Proposition 2.** *Assume that $\hat{\mathbf{g}}_t$ follows $\mathcal{N}(\mathbf{0}, \sigma^2\mathbf{I})$, we have $\mathbf{E}(\|\check{\mathbf{g}}_t\|^2) \leq ((2/\pi)^{\frac{1}{2}}c\Delta_g + 1)\mathbf{E}(\|\hat{\mathbf{g}}_t\|^2)$.*

However, the quantized clipped gradient may now be biased (i.e., $\mathbf{E}(\check{\mathbf{g}}_t) = \text{Clip}(\hat{\mathbf{g}}_t) \neq \hat{\mathbf{g}}_t$). The following Proposition shows that the bias is related to the clipping factor $c$. A larger $c$ (i.e., less severe gradient clipping) leads to smaller bias.

**Proposition 3.** *Assume that $\hat{\mathbf{g}}_t$ follows $\mathcal{N}(\mathbf{0}, \sigma^2\mathbf{I})$, we have $\mathbf{E}(\|Clip(\hat{\mathbf{g}}_t) - \hat{\mathbf{g}}_t\|^2) \leq d\sigma^2(2/\pi)^{\frac{1}{2}}F(c)$, where $F(c) = -ce^{-\frac{c^2}{2}} + \sqrt{\frac{\pi}{2}}(1+c^2)(1 - erf(\frac{c}{\sqrt{2}}))$, and $erf(z) = \frac{2}{\sqrt{\pi}}\int_0^z e^{-t^2}\,dt$ is the error function.*

**Theorem 3.** *Assume that $\hat{\mathbf{g}}_t$ follows $\mathcal{N}(\mathbf{0}, \sigma^2\mathbf{I})$. For loss-aware weight quantization with quantized clipped gradients and $\eta_t = \eta/\sqrt{t}$,*

$$\mathbf{E}(R(T)) \leq \frac{D_\infty^2 \sqrt{dT}}{2\eta}\sqrt{\sum_{t=1}^T (1-\beta)\beta^{T-t}\mathbf{E}(\|\check{\mathbf{g}}_t\|^2)} + \frac{\eta G_\infty \sqrt{d}}{\sqrt{1-\beta}}\sqrt{\sum_{t=1}^T \mathbf{E}(\|\check{\mathbf{g}}_t\|^2)}$$

$$+ \sqrt{L}D\sum_{t=1}^T \mathbf{E}(\sqrt{\|\mathbf{w}_t - \hat{\mathbf{w}}_t\|_{\mathbf{H}_t'}^2}) + D\sum_{t=1}^T \mathbf{E}(\sqrt{\|Clip(\hat{\mathbf{g}}_t) - \hat{\mathbf{g}}_t\|^2}), \quad (8)$$

$$\mathbf{E}\left(\frac{R(T)}{T}\right) \leq O\left(\sqrt{(2/\pi)^{\frac{1}{2}}c\Delta_g + 1}\frac{d}{\sqrt{T}}\right) + LD\sqrt{D^2 + \frac{d\alpha^2\Delta_w^2}{4}} + \sqrt{d}D\sigma(2/\pi)^{\frac{1}{4}}\sqrt{F(c)}. \quad (9)$$

Note that terms involving $\tilde{\mathbf{g}}_t$ in Theorem 2 are replaced by $\check{\mathbf{g}}_t$. Moreover, the regret has an additional term $D\sum_{t=1}^T \mathbf{E}(\sqrt{\|\text{Clip}(\hat{\mathbf{g}}_t) - \hat{\mathbf{g}}_t\|^2})$ over that in Theorem 2. Comparing the average regrets in Theorems 1 and 3, gradient clipping before quantization slows convergence by a factor of $\sqrt{(2/\pi)^{\frac{1}{2}}c\Delta_g + 1}$, as compared to using full-precision gradients. This is independent of $d$ as the increase in $\mathbf{E}(\|\check{\mathbf{g}}_t\|^2)$ is independent of $d$ (Proposition 2). Hence, a $\Delta_g$ larger than the one in Theorem 2 can be used, and this reduces the communication cost in distributed learning.

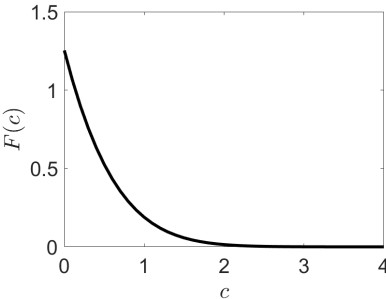

Figure 2: $F(c)$ vs clipping factor $c$.

A larger $c$ (i.e., less severe gradient clipping) makes $F(c)$ smaller (Figure 2). Compared with (6) in Theorem 2, the extra error $\sqrt{d}D\sigma(2/\pi)^{\frac{1}{4}}\sqrt{F(c)}$ in (9) is thus smaller, but convergence is also slower. Hence, there is a trade-off between the two.

**Remark 1.** *There are two scaling schemes in distributed training with data parallelism: strong scaling and weak scaling (Snavely et al., 2002). In this work, we consider weak scaling, which is more popular in deep network training. In weak scaling, the same data set size is used for each worker. The gradients are averaged over the $N$ workers as[1] $\mathbf{g}_t = \frac{1}{N}\sum_{n=1}^N \mathbf{g}_t^{(n)}$. If the gradients before averaging are independent random variables with zero mean, and $\|\mathbf{g}_t^{(n)}\|^2$ is bounded by $G^2$, then $\mathbf{E}(\|\mathbf{g}_t\|^2) = \mathbf{E}(\|\frac{1}{N}\sum_{n=1}^N \mathbf{g}_t^{(n)}\|^2) = \frac{1}{N^2}\mathbf{E}(\sum_{n=1}^N \|\mathbf{g}_t^{(n)}\|^2) \leq G^2/N$. From Theorems 1-3, the convergence speed with one worker is determined by $\frac{D_\infty^2\sqrt{dT}}{2\eta}\sqrt{\sum_{t=1}^T(1-\beta)\beta^{T-t}\mathbf{E}(\|\mathbf{g}_t\|^2)} + \frac{\eta G_\infty\sqrt{d}}{\sqrt{1-\beta}}\sqrt{\sum_{t=1}^T \mathbf{E}(\|\mathbf{g}_t\|^2)} \leq \frac{D_\infty^2\sqrt{d}}{2\eta}\sqrt{TG^2} + \frac{\eta G_\infty\sqrt{d}}{\sqrt{1-\beta}}\sqrt{TG^2} \leq (\frac{D_\infty^2\sqrt{d}}{2\eta} + \frac{\eta G_\infty\sqrt{d}}{\sqrt{1-\beta}})\sqrt{TG^2}$, while with $N$ workers by $(\frac{D_\infty^2\sqrt{d}}{2\eta} + \frac{\eta G_\infty\sqrt{d}}{\sqrt{1-\beta}})\sqrt{TG^2/N} = (\frac{D_\infty^2\sqrt{d}}{2\eta} + \frac{\eta G_\infty\sqrt{d}}{\sqrt{1-\beta}})\sqrt{(T/N)G^2}$. Thus, with $N$ workers, the number of iterations for convergence is subsequently reduced by a factor of $1/N$ as compared to using a single worker.*

## 4 EXPERIMENTS

### 4.1 SYNTHETIC DATA

In this section, we first study the effect of dimension $d$ on the convergence speed and final error of a simple linear model with square loss as in (Zhang et al., 2017). Each entry of the model parameter

---

[1]With a slight abuse of notation, $\mathbf{g}_t$ here can be the full-precision gradient or quantized gradient with/without clipping.

is generated by uniform sampling from $[-0.5, 0.5]$. Samples $\mathbf{x}_i$'s are generated such that each entry of $\mathbf{x}_i$ is drawn uniformly from $[-0.5, 0.5]$, and the corresponding output $y_i$ from $\mathcal{N}(\mathbf{x}_i^\top \mathbf{w}^*, (0.2)^2)$. At the $t$th iteration, a mini-batch of $B = 64$ samples are drawn to form $\mathbf{X}_t = [\mathbf{x}_1, \ldots, \mathbf{x}_B]$ and $\mathbf{y}_t = [y_1, \ldots, y_B]^\top$. The corresponding loss is $f_t(\mathbf{w}_t) = \|\mathbf{X}_t^\top \mathbf{w}_t - \mathbf{y}_t\|^2 / 2B$. The weights are quantized to 1 bit using LAB. The gradients are either full-precision (denoted FP) or stochastically quantized to 2 bits (denoted SQ2). The optimizer is RMSProp, and the learning rate is $\eta_t = \eta / \sqrt{t}$, where $\eta = 0.03$. Training is terminated when the average training loss does not decrease for 5000 iterations.

Figure 3(a) shows[2] convergence of the average training loss $\sum_{t=1}^T f_t(\mathbf{w}_t)/T$, which differs from the average regret only by only a constant. As can be seen, for both full-precision and quantized gradients, a larger $d$ leads to a larger loss upon convergence. Moreover, convergence is slower for larger $d$, particularly when the gradients are quantized. These agree with the results in Theorems 1 and 2.

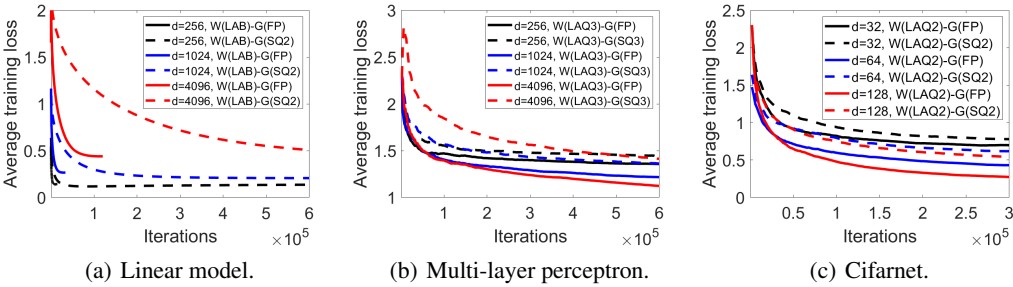

(a) Linear model.  (b) Multi-layer perceptron.  (c) Cifarnet.

Figure 3: Convergence of the weight-quantized models with different $d$'s.

## 4.2 CIFAR-10

In this experiment, we follow (Wen et al., 2017) and use the same train/test split, data preprocessing, augmentation and distributed Tensorflow setup.

### 4.2.1 VARYING $d$

We first study the effect of $d$ on deep networks. Experiments are performed on two neural network models. The first one is a multi-layer perceptron with one layer of $d$ hidden units (Reddi et al., 2016). The weights are quantized to 3 bits using LAQ3. The gradients are either full-precision (denoted FP) or stochastically quantized to 3 bits (denoted SQ3). The optimizer is RMSProp, and the learning rate is $\eta_t = \eta / \sqrt{t}$, where $\eta = 0.1$. The second network is the Cifarnet (Wen et al., 2017). We set $d$ to be the number of filters in each convolutional layer. The gradients are either full-precision or stochastically quantized to 2 bits (denoted SQ2). Adam is used as the optimizer. The learning rate is decayed from 0.0002 by a factor of 0.1 every 200 epochs as in (Wen et al., 2017).

Figures 3(b) and 3(c) show convergence of the average training loss for both networks. As can be seen, similar to that in Section 4.1, a larger $d$ leads to larger convergence degradation of the quantized gradients as compared to using full-precision gradients. However, unlike the linear model, a larger $d$ does not necessarily lead to a larger loss upon convergence.

### 4.2.2 WEIGHT QUANTIZATION RESOLUTION $\Delta_w$

We use the same Cifarnet model as in (Wen et al., 2017), with $d = 64$. Weights are quantized to 1 bit (LAB), 2 bits (LAQ2), or $m$ bits (LAQ$m$). The gradients are full-precision (FP) or stochastically quantized to $m = \{2, 3, 4\}$ bits (SQ$m$) without gradient clipping. Adam is used as the optimizer. The learning rate is decayed from 0.0002 by a factor of 0.1 every 200 epochs. Two workers are used in this experiment. Figure 4 shows convergence of the average training loss with different numbers of bits for the quantized weight. With full-precision or quantized gradients, weight-quantized networks have larger training losses than full-precision networks upon convergence. The more bits are used, the smaller is the final loss. This agrees with the results in Theorems 1 and 2. Table 1 shows the test

---

[2]Legend "W(LAB)-G(FP)" means that weights are quantized using LAB and gradients are full-precision.

set accuracies. Weight-quantized networks are less accurate than their full-precision counterparts, but the degradation is small when 3 or 4 bits are used.

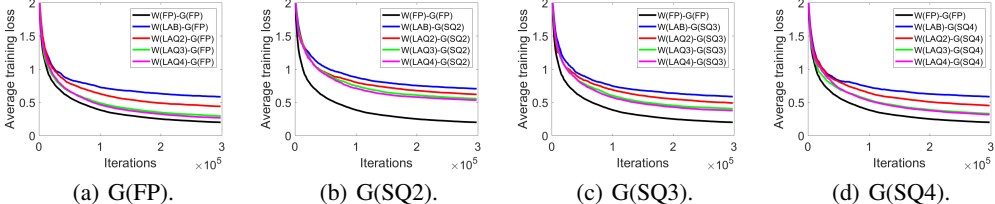

(a) G(FP).  (b) G(SQ2).  (c) G(SQ3).  (d) G(SQ4).

Figure 4: Convergence with different numbers of bits for the weights on CIFAR-10. The gradient is full-precision (denoted G(FP)) or $m$-bit quantized (denoted G(SQ$m$)) without gradient clipping.

Table 1: Testing accuracy (%) on CIFAR-10 with two workers.

| gradient \ weight | FP | LAB | LAQ2 | LAQ3 | LAQ4 |
|---|---|---|---|---|---|
| FP | 83.74 | 80.37 | 82.11 | 83.14 | 83.35 |
| SQ2 (no clipping) | 81.40 | 78.67 | 80.27 | 81.27 | 81.38 |
| SQ2 (clip,$c = 3$) | 82.99 | 80.25 | 81.59 | 83.14 | 83.40 |
| SQ3 (no clipping) | 83.24 | 80.18 | 81.63 | 82.75 | 83.17 |
| SQ3 (clip, $c = 3$) | 83.89 | 80.13 | 81.77 | 82.97 | 83.43 |
| SQ4 (no clipping) | 83.64 | 80.44 | 81.88 | 83.13 | 83.47 |
| SQ4 (clip, $c = 3$) | 83.80 | 79.27 | 81.42 | 82.77 | 83.43 |

### 4.2.3 GRADIENT QUANTIZATION RESOLUTION $\Delta_g$

We use the same Cifarnet model as in (Wen et al., 2017). Adam is used as the optimizer. The learning rate is decayed from 0.0002 by a factor of 0.1 every 200 epochs. Figure 5 shows convergence of the average training loss with different numbers of bits for the quantized gradients, again without gradient clipping. Using fewer bits yields a larger final error, and using 2- or 3-bit gradients yields larger training loss and worse accuracy than full-precision gradients (Figure 5 and Table 1). The fewer bits for the gradients, the larger the gap. The degradation is negligible when 4 bits are used. Indeed, 4-bit gradient sometimes has even better accuracy than full-precision gradient, as its inherent randomness encourages escape from poor sharp minima (Wen et al., 2017). Moreover, using a larger $m$ results in faster convergence, which agrees with Theorem 2.

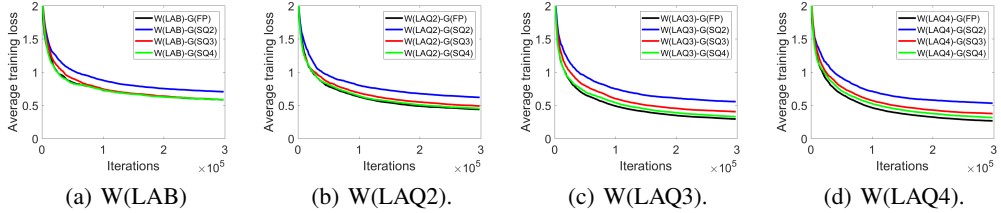

(a) W(LAB)  (b) W(LAQ2).  (c) W(LAQ3).  (d) W(LAQ4).

Figure 5: Convergence with different numbers of bits for the gradients on CIFAR-10. The weight is binarized (denoted W(LAB)) or $m$-bit quantized (denoted W(LAQ$m$)). Gradients are not clipped.

### 4.2.4 GRADIENT CLIPPING

In this section, we perform experiments on gradient clipping, with clipping factor $c$ in $\{1, 2, 3\}$, using the Cifarnet (Wen et al., 2017). LAQ2 is used for weight quantization and SQ2 for gradient quantization. Adam is used as the optimizer. The learning rate is decayed from 0.0002 by a factor of 0.1 every 200 epochs. Figure 6(a) shows histograms of the full-precision gradients before clipping. As can be seen, the gradients at each layer before clipping roughly follow the normal distribution, which verifies the assumption in Section 3.4. Figure 6(b) shows the average $\|\tilde{\mathbf{g}}_t\|^2/\|\hat{\mathbf{g}}_t\|^2$ (for non-clipped gradients) and $\|\check{\mathbf{g}}_t\|^2/\|\hat{\mathbf{g}}_t\|^2$ (for clipped gradients) over all iterations. The dimensionalities ($d$) of the various Cifarnet layers are "conv1": 1600, "conv2": 1600, "fc3": 884736, "fc4": 73728,

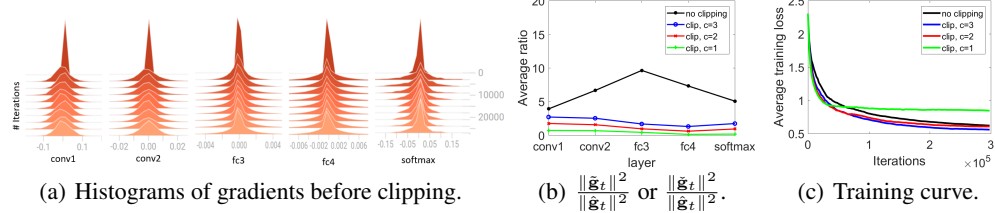

(a) Histograms of gradients before clipping.

(b) $\frac{\|\tilde{\mathbf{g}}_t\|^2}{\|\hat{\mathbf{g}}_t\|^2}$ or $\frac{\|\check{\mathbf{g}}_t\|^2}{\|\hat{\mathbf{g}}_t\|^2}$.

(c) Training curve.

Figure 6: Results for LAQ2 with SQ2 on CIFAR-10 with two workers. (a) Histograms of gradients at different Cifarnet layers before clipping (visualized by Tensorboard); (b) Average $\|\tilde{\mathbf{g}}_t\|^2/\|\hat{\mathbf{g}}_t\|^2$ (for non-clipped gradients) and $\|\check{\mathbf{g}}_t\|^2/\|\hat{\mathbf{g}}_t\|^2$ (for clipped gradients); and (c) Training curves.

"softmax": 1920. Layers with large $d$ have large $\|\tilde{\mathbf{g}}_t\|^2/\|\hat{\mathbf{g}}_t\|^2$ values, which agrees with Proposition 1. With clipped gradients, $\|\check{\mathbf{g}}_t\|^2/\|\hat{\mathbf{g}}_t\|^2$ is much smaller and does not depend on $d$, agreeing with Proposition 3. Figure 6(c) shows convergence of the average training loss. Using a smaller $c$ (more aggressive clipping) leads to faster training (at the early stage of training) but larger final training loss, agreeing with Theorem 3.

Figure 7 shows convergence of the average training loss with different numbers of bits for the quantized clipped gradient, with $c = 3$. By comparing[3] with Figure 5, gradient clipping achieves faster convergence, especially when the number of gradient bits is small. For example, 2-bit clipped gradient has comparable speed (Figure 7) and accuracy (Table 1) as full-precision gradient.

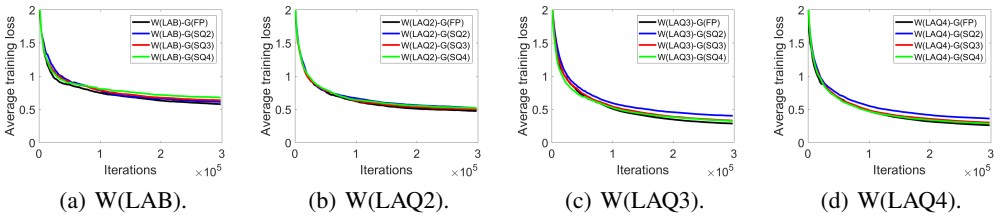

(a) W(LAB).     (b) W(LAQ2).     (c) W(LAQ3).     (d) W(LAQ4).

Figure 7: Convergence with different numbers of bits for gradients (with $c = 3$) on CIFAR-10.

### 4.2.5 VARYING THE NUMBER OF WORKERS

In Remark 1, we showed that using multiple workers can reduce the number of training iterations required. In this section, we vary the number of workers in a distributed learning setting with weak scaling, using the Cifarnet (Wen et al., 2017). We fix the mini-batch size for each worker to 64, and set a smaller number of iterations when more workers are used.

We use 3-bit quantized weight (LAQ3), and gradients are full-precision or stochastically quantized to $m = \{2, 3, 4\}$ bits (SQ$m$). Table 2 shows the testing accuracies with varying number of workers $N$. Observations are similar to those in Section 4.2.4. 2-bit quantized clipped gradient has comparable performance as full-precision gradient, while the non-clipped counterpart requires 3 to 4 bits for comparable performance.

Table 2: Testing accuracy (%) on CIFAR-10 with varying number of workers ($N$).

| weight | gradient | $N = 4$ | $N = 8$ | $N = 16$ |
|--------|----------|---------|---------|----------|
| FP | FP | 83.28 | 83.38 | 83.76 |
| | FP | 82.92 | 82.93 | 83.12 |
| | SQ2 (no clipping) | 81.53 | 81.08 | 81.30 |
| | SQ2 (clip, $c = 3$) | 83.01 | 82.94 | 82.73 |
| LAQ3 | SQ3 (no clipping) | 82.64 | 82.42 | 82.27 |
| | SQ3 (clip, $c = 3$) | 82.93 | 83.16 | 82.54 |
| | SQ4 (no clipping) | 83.03 | 82.53 | 82.61 |
| | SQ4 (clip, $c = 3$) | 82.51 | 83.07 | 82.52 |

---

[3]The curves for full-precision gradient are the same in Figures 5 and 7, and can be used as a common baseline.

### 4.3 IMAGENET

In this section, we train the AlexNet on ImageNet. We follow (Wen et al., 2017) and use the same data preprocessing, augmentation, learning rate, and mini-batch size. Quantization is not performed in the first and last layers, as is common in the literature (Zhou et al., 2016; Zhu et al., 2017; Polino et al., 2018; Wen et al., 2017). We use Adam as the optimizer. We experiment with 4-bit loss-aware weight quantization (LAQ4), and the gradients are either full-precision or quantized to 3 bits (SQ3).

Table 3 shows the accuracies with different numbers of workers. Weight-quantized networks have slightly worse accuracies than full-precision networks. Quantized clipped gradient outperforms the non-clipped counterpart, and achieves comparable accuracy as full-precision gradient.

Table 3: Top-1 and top-5 accuracies (%) on ImageNet.

| weight | gradient | $N = 2$ | | $N = 4$ | | $N = 8$ | |
|---|---|---|---|---|---|---|---|
| | | top-1 | top-5 | top-1 | top-5 | top-1 | top-5 |
| FP | FP | 55.08 | 78.33 | 55.45 | 78.57 | 55.40 | 78.69 |
| | FP | 53.79 | 77.21 | 54.22 | 77.53 | 54.73 | 78.12 |
| LAQ4 | SQ3 (no clipping) | 52.48 | 75.97 | 52.87 | 76.40 | 53.18 | 76.62 |
| | SQ3 (clip, $c = 3$) | 54.13 | 77.27 | 54.23 | 77.55 | 54.34 | 78.07 |

Figure 8 shows the speedup in distributed training of a weight-quantized network with quantized/full-precision gradient compared to training with one worker using full-precision gradient. We use the performance model in (Wen et al., 2017), which combines lightweight profiling on a single node with analytical communication modeling. We use the AllReduce communication model (Rabenseifner, 2004), in which each GPU communicates with its neighbor until all gradients are accumulated to a single GPU. We do not include the server's computation effort on weight quantization and the worker's effort on gradient clipping, which are negligible compared to the forward and backward propagations in the worker. As can be seen from the Figure, even though the number of bits used for gradients increases by one at every aggregation step in the AllReduce model, the proposed method still significantly reduces network communication and speeds up training. When the bandwidth is small (Figure 8(a)), communication is the bottleneck, and using quantizing gradients is significantly faster than the use of full-precision gradients. With a larger bandwidth (Figure 8(b)), the difference in speedups is smaller. Moreover, note that on the 1Gbps Ethernet with quantized gradients, its speedup is similar to those on the 10Gbps Ethernet with full-precision gradients.

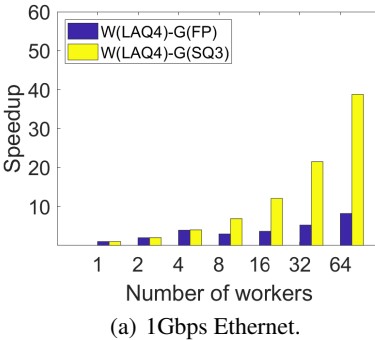
(a) 1Gbps Ethernet.

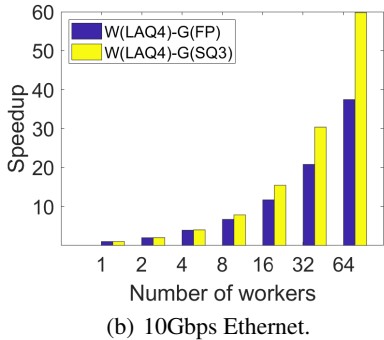
(b) 10Gbps Ethernet.

Figure 8: Speedup of ImageNet training on a 16-node GPU cluster. Each node has 4 1080ti GPUs with one PCI switch.

## 5 CONCLUSION

In this paper, we studied loss-aware weight-quantized networks with quantized gradient for efficient communication in a distributed environment. Convergence analysis is provided for weight-quantized models with full-precision, quantized and quantized clipped gradients. Empirical experiments confirm the theoretical results, and demonstrate that quantized networks can speed up training and have comparable performance as full-precision networks.

ACKNOWLEDGMENTS

We thank NVIDIA for the gift of GPU card.

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

## A  PROOFS

### A.1  PROOF OF THEOREM 1

First, we introduce the following two lemmas.

**Lemma 1.** *Let $\mathbf{v}_t = \sum_{j=1}^{t}(1-\beta)\beta^{t-j}\mathbf{g}_j^2$, where $\beta \in [0,1)$. Assume that $\|\mathbf{g}_j\|_\infty < G_\infty$ for $j \in \{1, 2 \ldots, t\}$. Then,*

$$v_{t,i} \geq (1-\beta)g_{t,i}^2, \tag{10}$$
$$\sqrt{v_{t,i}} \leq G_\infty, \tag{11}$$

$$\|\sqrt{\mathbf{v}_t}\| = \sqrt{\sum_{j=1}^{t}(1-\beta)\beta^{t-j}\|\mathbf{g}_j\|^2}. \tag{12}$$

*Proof.* $v_{t,i} = \sum_{j=1}^{t}(1-\beta)\beta^{t-j}g_{j,i}^2 \geq (1-\beta)g_{t,i}^2$. Moreover,

$$\sqrt{v_{t,i}} = \sqrt{\sum_{j=1}^{t}(1-\beta)\beta^{t-j}g_{j,i}^2} \leq \sqrt{\sum_{j=1}^{t}(1-\beta)\beta^{t-j}G_\infty^2} \leq G_\infty,$$

and

$$\|\sqrt{\mathbf{v}_t}\| = \sqrt{\sum_{i=1}^{d}v_{t,i}} = \sqrt{\sum_{i=1}^{d}\sum_{j=1}^{t}(1-\beta)\beta^{t-j}g_{j,i}^2} = \sqrt{\sum_{j=1}^{t}(1-\beta)\beta^{t-j}\|\mathbf{g}_j\|^2}.$$

$\square$

**Lemma 2.** *[Lemma 10.3 in (Kingma & Ba, 2015)] Let $\mathbf{g}_{1:T,i} = [g_{1,i}, g_{2,i}, \ldots, g_{T,i}]^\top$ be the vector containing the $i$th element of the gradients for all iterations up to $T$, and $\mathbf{g}_t$ be bounded as in Assumption **A3**. Then,*

$$\sum_{t=1}^{T}\sqrt{\frac{g_{t,i}^2}{t}} \leq 2G_\infty\|\mathbf{g}_{1:T,i}\|.$$

*Proof.* (of Theorem 1) When only weights are quantized, the update for loss-aware weight quantization is

$$\mathbf{w}_{t+1} = \mathbf{w}_t - \eta_t\mathrm{Diag}(\sqrt{\hat{\mathbf{v}}_t})^{-1}\hat{\mathbf{g}}_t. \tag{13}$$

The update for the $i$th entry of $\mathbf{w}_t$ is $w_{t+1,i} = w_{t,i} - \eta_t\frac{\hat{g}_{t,i}}{\sqrt{\hat{v}_{t,i}}}$. This implies

$$(w_{t+1,i} - w_i^*)^2 = (w_{t,i} - w_i^*)^2 - 2\eta_t(w_{t,i} - w_i^*)\frac{g_{t,i}}{\sqrt{\hat{v}_{t,i}}}$$
$$+2\eta_t(w_{t,i} - w_i^*)\frac{(g_{t,i} - \hat{g}_{t,i})}{\sqrt{\hat{v}_{t,i}}} + \eta_t^2(\frac{\hat{g}_{t,i}}{\sqrt{\hat{v}_{t,i}}})^2.$$

After rearranging,

$$g_{t,i}(w_{t,i} - w_i^*) = \frac{\sqrt{\hat{v}_{t,i}}}{2\eta_t}((w_{t,i} - w_i^*)^2 - (w_{t+1,i} - w_i^*)^2)$$
$$+(w_{t,i} - w_i^*)(g_{t,i} - \hat{g}_{t,i}) + \frac{\eta_t}{2}(\frac{\hat{g}_{t,i}^2}{\sqrt{\hat{v}_{t,i}}}). \tag{14}$$

Since $f_t$ is convex, we have

$$f_t(\mathbf{w}_t) - f_t(\mathbf{w}^*) \leq \mathbf{g}_t^\top(\mathbf{w}_t - \mathbf{w}^*) = \sum_{i=1}^{d}g_{t,i}(w_{t,i} - w_i^*). \tag{15}$$

As $f_t$ is convex (Assumption **A1**) and twice differentiable (Assumption **A2**), $\nabla^2 f_t \preceq L\mathbf{I}$, where $\mathbf{I}$ is the identity matrix. Combining with the assumption in Section 3.1 that $\|\mathbf{w}_m - \mathbf{w}_n\| \leq D$, we have

$$\|\mathbf{w}_t - \mathbf{w}^*\|^2_{\mathbf{H}'_t} \leq L\|\mathbf{w}_t - \mathbf{w}^*\|^2 \leq LD^2, \quad \|\mathbf{w}_t - \hat{\mathbf{w}}_t\|^2_{\mathbf{H}'_t} \leq L\|\mathbf{w}_t - \hat{\mathbf{w}}_t\|^2 \leq LD^2. \tag{16}$$

Let $\langle \mathbf{x}, \mathbf{y} \rangle = \mathbf{x}^\top \mathbf{y}$ be the dot product between two vectors $\mathbf{x}$ and $\mathbf{y}$. Combining (14) and (15), sum over all the dimensions $i \in \{1, 2 \ldots, d\}$ and over all iterations $t \in \{1, 2, \ldots, T\}$, we have

$$
\begin{aligned}
R(T) &= \sum_{t=1}^T f_t(\mathbf{w}_t) - f_t(\mathbf{w}^*) \\
&\leq \sum_{i=1}^d \frac{\sqrt{\hat{v}_{1,i}}}{2\eta_1}(w_{1,i} - w_i^*)^2 + \sum_{i=1}^d \sum_{t=2}^T \left( \frac{\sqrt{\hat{v}_{t,i}}}{2\eta_t} - \frac{\sqrt{\hat{v}_{t-1,i}}}{2\eta_{t-1}} \right)(w_{t,i} - w_i^*)^2 \\
&\quad + \sum_{t=1}^T \langle \mathbf{w}_t - \mathbf{w}^*, \mathbf{g}_t - \hat{\mathbf{g}}_t \rangle + \sum_{i=1}^d \sum_{t=1}^T \frac{\eta_t}{2} \left( \frac{\hat{g}^2_{t,i}}{\sqrt{(1-\beta)\hat{g}^2_{t,i}}} \right) \\
&\leq \frac{D^2_\infty}{2\eta} \sum_{i=1}^d \sqrt{T\hat{v}_{T,i}} + \sum_{t=1}^T \langle \mathbf{H}'^{\frac{1}{2}}_t (\mathbf{w}_t - \mathbf{w}^*), \mathbf{H}'^{-\frac{1}{2}}_t (\mathbf{g}_t - \hat{\mathbf{g}}_t) \rangle + \frac{\eta G_\infty}{\sqrt{1-\beta}} \sum_{i=1}^d \|\hat{\mathbf{g}}_{1:T,i}\| \\
&\leq \frac{D^2_\infty}{2\eta} \sum_{i=1}^d \sqrt{T\hat{v}_{T,i}} + \sum_{t=1}^T \sqrt{\|\mathbf{w}_t - \mathbf{w}^*\|^2_{\mathbf{H}'_t}} \sqrt{\|\mathbf{w}_t - \hat{\mathbf{w}}_t\|^2_{\mathbf{H}'_t}} + \frac{\eta G_\infty}{\sqrt{1-\beta}} \sum_{i=1}^d \|\hat{\mathbf{g}}_{1:T,i}\| \\
&\leq \frac{D^2_\infty}{2\eta} \sum_{i=1}^d \sqrt{T\hat{v}_{T,i}} + \frac{\eta G_\infty}{\sqrt{1-\beta}} \sum_{i=1}^d \|\hat{\mathbf{g}}_{1:T,i}\| + \sqrt{L}D \sum_{t=1}^T \sqrt{\|\mathbf{w}_t - \hat{\mathbf{w}}_t\|^2_{\mathbf{H}'_t}} \\
&\leq \frac{D^2_\infty}{2\eta} \sqrt{dT} \sqrt{\sum_{t=1}^T (1-\beta)\beta^{T-t}\|\hat{\mathbf{g}}_t\|^2} + \frac{\eta G_\infty}{\sqrt{1-\beta}} \sqrt{d} \sqrt{\sum_{t=1}^T \|\hat{\mathbf{g}}_t\|^2} \\
&\quad + \sqrt{L}D \sum_{t=1}^T \sqrt{\|\mathbf{w}_t - \hat{\mathbf{w}}_t\|^2_{\mathbf{H}'_t}}.
\end{aligned}
\tag{17}
$$

The first inequality comes from (10) in Lemma 1. In the second inequality, the first term comes from $\sum_{i=1}^d \frac{\sqrt{\hat{v}_{1,i}}}{2\eta_1}(w_{1,i} - w_i^*)^2 + \sum_{i=1}^d \sum_{t=2}^T \left( \frac{\sqrt{\hat{v}_{t,i}}}{2\eta_t} - \frac{\sqrt{\hat{v}_{t-1,i}}}{2\eta_{t-1}} \right)(w_{t,i} - w_i^*)^2 = \sum_{i=1}^d \frac{\sqrt{\hat{v}_{T,i}}}{2\eta_T}(w_{T,i} - w_i^*)^2$ and the domain bound assumption in Section 3.1 (i.e., $(w_{T,i} - w_i^*) \leq \|\mathbf{w}_T - \mathbf{w}^*\|_\infty \leq D_\infty$). The second term comes from the definition of $\mathbf{H}'_t$ (i.e., $\mathbf{H}'^{-\frac{1}{2}}_t (\mathbf{g}_t - \hat{\mathbf{g}}_t) = \mathbf{H}'^{\frac{1}{2}}_t (\mathbf{w}_t - \hat{\mathbf{w}}_t)$), and Lemma 2. The third inequality comes from Cauchy's inequality. The fourth inequality comes from (16). The last inequality comes from (12) in Lemma 1.

For $m$-bit ($m > 1$) loss-aware weight quantization in (13), as

$$\hat{\mathbf{w}}_t = \arg \quad \min_{\mathbf{w}} \quad \|\hat{\mathbf{w}} - \mathbf{w}_t\|^2_{\text{Diag}(\sqrt{\hat{\mathbf{v}}_{t-1}})} = \sum_{i=1}^d \sqrt{\hat{v}_{t-1,i}}(\hat{w}_i - w_{t,i})^2$$

$$\text{s.t.} \quad \hat{\mathbf{w}} = \alpha_t \mathbf{b}_t, \ \alpha > 0, \ \mathbf{b} \in (\mathcal{S}_w)^d.$$

If $-\alpha_t M_k \leq w_{t,i} \leq \alpha_t M_k$, as $\sqrt{\hat{v}_{t-1,i}} > 0$, the optimal $\hat{w}_{t,i}$ satisfies $\hat{w}_{t,i} \in \{\alpha_t M_{r,i}, \alpha_t M_{r+1,i}\}$, where $r$ is the index that satisfies $\alpha_t M_{r,i} \leq w_{t,i} \leq \alpha_t M_{r+1,i}$. Since $\alpha = \max\{\alpha_1, \ldots, \alpha_T\}$, we have

$$
\begin{aligned}
(\hat{w}_{t,i} - w_{t,i})^2 &= \min\{(\alpha_t M_{r+1,i} - |w_{t,i}|)^2, (\alpha_t M_{r,i} - |w_{t,i}|)^2\} \\
&\leq \left( \frac{\alpha_t M_{r+1,i} - \alpha_t M_{r,i}}{2} \right)^2 \leq \frac{\alpha^2 \Delta^2_w}{4}.
\end{aligned}
\tag{18}
$$

Otherwise (i.e., $w_{t,i}$ is exterior of the representable range), the optimal $\hat{w}_{t,i}$ is just the nearest representable value of $w_{t,i}$. Thus,

$$(\hat{w}_{t,i} - w_{t,i})^2 = (|w_{t,i}| - \alpha_t M_k)^2 \leq w^2_{t,i}. \tag{19}$$

From (18) and (19), and sum over all the dimensions, we have

$$\|\hat{\mathbf{w}}_t - \mathbf{w}_t\|^2 \leq \frac{d\alpha^2 \Delta_w^2}{4} + \|\mathbf{w}_t\|^2 \leq \frac{d\alpha^2 \Delta_w^2}{4} + D^2. \tag{20}$$

From (16) and (20),

$$\|\hat{\mathbf{w}}_t - \mathbf{w}_t\|^2_{\mathbf{H}'_t} \leq L\left(D^2 + \frac{d\alpha^2 \Delta_w^2}{4}\right). \tag{21}$$

From (21) and Assumption **A3**, we have from (17)

$$R(T) \leq \frac{D^2_\infty dG_\infty \sqrt{T}}{2\eta} + \frac{\eta dG^2_\infty \sqrt{T}}{\sqrt{1-\beta}} + LD\sqrt{D^2 + \frac{d\alpha^2 \Delta_w^2}{4}}T.$$

Thus, the average regret is

$$R(T)/T \leq \left(\frac{D^2_\infty G_\infty}{2\eta} + \frac{\eta G^2_\infty}{\sqrt{1-\beta}}\right)\frac{d}{\sqrt{T}} + LD\sqrt{D^2 + \frac{d\alpha^2 \Delta_w^2}{4}}. \tag{22}$$

$\square$

### A.2 PROOF OF PROPOSITION 1

**Lemma 3.** *For stochastic gradient quantization in (3), $\mathbf{E}(\tilde{\mathbf{g}}_t) = \hat{\mathbf{g}}_t$, and $\mathbf{E}(\|\tilde{\mathbf{g}}_t - \hat{\mathbf{g}}_t\|^2) \leq \Delta_g\|\hat{\mathbf{g}}_t\|_\infty\|\hat{\mathbf{g}}_t\|_1$.*

*Proof.* Denote the $i$th element of the quantized gradient $\tilde{\mathbf{g}}_t$ by $\tilde{g}_{t,i}$. For two adjacent quantized values $B_{r,i}, B_{r+1,i}$ with $B_{r,i} \leq |g_{t,i}|/s_t < B_{r+1,i}$,

$$\begin{aligned}
\mathbf{E}(\tilde{g}_{t,i}) &= \mathbf{E}(s_t \cdot \text{sign}(\hat{g}_{t,i}) \cdot q_{t,i}) = s_t \cdot \text{sign}(\hat{g}_{t,i}) \cdot \mathbf{E}(q_{t,i}) \\
&= s_t \cdot \text{sign}(\hat{g}_{t,i}) \cdot (pB_{r+1,i} + (1-p)B_{r,i}) \\
&= s_t \cdot \text{sign}(\hat{g}_{t,i}) \cdot (p(B_{r+1,i} - B_{r,i}) + B_{r,i}) \\
&= s_t \cdot \text{sign}(\hat{g}_{t,i}) \cdot \frac{|\hat{g}_{t,i}|}{s_t} = \hat{g}_{t,i}.
\end{aligned}$$

Thus, $\mathbf{E}(\tilde{\mathbf{g}}_t) = \hat{\mathbf{g}}_t$, and the variance of the quantized gradients satisfy

$$\begin{aligned}
\mathbf{E}(\|\tilde{\mathbf{g}}_t - \hat{\mathbf{g}}_t\|^2) &= \sum_{i=1}^d (s_t B_{r+1,i} - |\hat{g}_{t,i}|)(|\hat{g}_{t,i}| - s_t B_{r,i}) \\
&= s_t^2 \sum_{i=1}^d (B_{r+1,i} - \frac{|\hat{g}_{t,i}|}{s_t})(\frac{|\hat{g}_{t,i}|}{s_t} - B_{r,i}) \\
&\leq s_t^2 \sum_{i=1}^d (B_{r+1,i} - B_{r,i})\frac{|\hat{g}_{t,i}|}{s_t} \\
&= \Delta_g\|\hat{\mathbf{g}}_t\|_\infty\|\hat{\mathbf{g}}_t\|_1.
\end{aligned}$$

$\square$

*Proof.* From Lemma 3,

$$\mathbf{E}(\|\tilde{\mathbf{g}}_t\|^2) = \mathbf{E}(\|\tilde{\mathbf{g}}_t - \hat{\mathbf{g}}_t\|^2) + \|\hat{\mathbf{g}}_t\|^2 \leq \Delta_g\|\hat{\mathbf{g}}_t\|_\infty\|\hat{\mathbf{g}}_t\|_1 + \|\hat{\mathbf{g}}_t\|^2.$$

Denote $\{x_1, x_2, \ldots, x_d\}$ as the absolute values of the elements in $\hat{\mathbf{g}}_t$ sorted in ascending order. From Cauchy's inequality, we have

$$\begin{aligned}
\|\hat{\mathbf{g}}_t\|_\infty\|\hat{\mathbf{g}}\|_1 &= x_d \sum_{i=1}^d x_i = \sum_{i=1}^d x_d x_i \leq \sum_{i=1}^{d-1}\left(\frac{1+\sqrt{2d-1}}{2}x_i^2 + \frac{1}{1+\sqrt{2d-1}}x_d^2\right) + x_d^2 \\
&= \frac{1+\sqrt{2d-1}}{2}\sum_{i=1}^d x_i^2 = \frac{1+\sqrt{2d-1}}{2}\|\hat{\mathbf{g}}_t\|^2.
\end{aligned}$$

The equality holds iff $x_1 = x_2 = \cdots = \frac{\sqrt{2}}{1+\sqrt{2d-1}}x_d$. Thus, we have

$$\mathbf{E}(\|\tilde{\mathbf{g}}_t\|^2) \leq (\frac{1+\sqrt{2d-1}}{2}\Delta_g + 1)\|\hat{\mathbf{g}}_t\|^2.$$

$\square$

### A.3 Proof of Theorem 2

*Proof.* When both weights and gradients are quantized, the update is

$$\mathbf{w}_{t+1} = \mathbf{w}_t - \eta_t \text{Diag}(\sqrt{\tilde{\mathbf{v}}_t})^{-1}\tilde{\mathbf{g}}_t. \tag{23}$$

Similar to the proof of Theorem 1, and using that $\mathbf{E}(\tilde{\mathbf{g}}_t) = \hat{\mathbf{g}}_t$ (Lemma 3), we have

$$
\begin{aligned}
\mathbf{E}(R(T)) &\leq \sum_{i=1}^{d}\mathbf{E}(\frac{\sqrt{\tilde{v}_{1,i}}}{2\eta_1}(w_{1,i}-w_i^*)^2) + \sum_{i=1}^{d}\sum_{t=2}^{T}\mathbf{E}\left((\frac{\sqrt{\tilde{v}_{t,i}}}{2\eta_t}-\frac{\sqrt{\tilde{v}_{t-1,i}}}{2\eta_{t-1}})(w_{t,i}-w_i^*)^2\right) \\
&\quad + \sum_{t=1}^{T}\mathbf{E}(\langle\mathbf{w}_t-\mathbf{w}^*, \mathbf{g}_t-\tilde{\mathbf{g}}_t\rangle) + \sum_{i=1}^{d}\sum_{t=1}^{T}\frac{\eta_t}{2}\mathbf{E}\left(\frac{\tilde{g}_{t,i}^2}{\sqrt{(1-\beta)\tilde{g}_{t,i}^2}}\right) \\
&\leq \frac{D_\infty^2}{2\eta}\sum_{i=1}^{d}\mathbf{E}(\sqrt{T\tilde{v}_{T,i}}) + \sum_{t=1}^{T}\mathbf{E}\langle\mathbf{H}_t^{'\frac{1}{2}}(\mathbf{w}_t-\mathbf{w}^*), \mathbf{H}_t^{'-\frac{1}{2}}(\mathbf{g}_t-\hat{\mathbf{g}}_t)\rangle \\
&\quad + \frac{\eta G_\infty}{\sqrt{1-\beta}}\sum_{i=1}^{d}\mathbf{E}(\|\tilde{\mathbf{g}}_{1:T,i}\|) \\
&\leq \frac{D_\infty^2}{2\eta}\sum_{i=1}^{d}\mathbf{E}(\sqrt{T\tilde{v}_{T,i}}) + \sum_{t=1}^{T}\mathbf{E}(\sqrt{\|\mathbf{w}_t-\mathbf{w}^*\|_{\mathbf{H}_t'}^2}\sqrt{\|\mathbf{w}_t-\hat{\mathbf{w}}_t\|_{\mathbf{H}_t'}^2}) \\
&\quad + \frac{\eta G_\infty}{\sqrt{1-\beta}}\sum_{i=1}^{d}\mathbf{E}(\|\tilde{\mathbf{g}}_{1:T,i}\|) \\
&\leq \frac{D_\infty^2}{2\eta}\sqrt{dT}\mathbf{E}(\|\sqrt{\tilde{\mathbf{v}}_T}\|) + \frac{\eta G_\infty}{\sqrt{1-\beta}}\sum_{i=1}^{d}\mathbf{E}(\|\tilde{\mathbf{g}}_{1:T,i}\|) \\
&\quad + \sqrt{L}D\sum_{t=1}^{T}\mathbf{E}(\sqrt{\|\mathbf{w}_t-\hat{\mathbf{w}}_t\|_{\mathbf{H}_t'}^2}) \\
&\leq \frac{D_\infty^2}{2\eta}\sqrt{dT}\sqrt{\sum_{t=1}^{T}(1-\beta)\beta^{T-t}\mathbf{E}(\|\tilde{\mathbf{g}}_t\|^2)} + \frac{\eta G_\infty}{\sqrt{1-\beta}}\sqrt{d}\sqrt{\sum_{t=1}^{T}\mathbf{E}(\|\tilde{\mathbf{g}}_t\|^2)} \\
&\quad + \sqrt{L}D\sum_{t=1}^{T}\mathbf{E}(\sqrt{\|\mathbf{w}_t-\hat{\mathbf{w}}_t\|_{\mathbf{H}_t'}^2}).
\end{aligned}
$$

As (21) in the proof of Theorem 1 still holds, using Proposition 1 and Assumption **A3**, we have

$$
\begin{aligned}
\mathbf{E}(R(T)) &\leq \left(\frac{D_\infty^2}{2\eta}\sqrt{dT}\sqrt{\sum_{t=1}^{T}(1-\beta)\beta^{T-t}\|\hat{\mathbf{g}}_t\|^2} + \frac{\eta G_\infty}{\sqrt{1-\beta}}\sqrt{d}\sqrt{\sum_{t=1}^{T}\|\hat{\mathbf{g}}_t\|^2}\right) \\
&\quad \cdot\sqrt{\frac{1+\sqrt{2d-1}}{2}\Delta_g + 1} + LD\sqrt{D^2 + \frac{d\alpha^2\Delta_w^2}{4}} \\
&\leq \left(\frac{D_\infty^2 G_\infty}{2\eta} + \frac{\eta G_\infty^2}{\sqrt{1-\beta}}\right)d\sqrt{T}\sqrt{\frac{1+\sqrt{2d-1}}{2}\Delta_g + 1} + LD\sqrt{D^2 + \frac{d\alpha^2\Delta_w^2}{4}}.
\end{aligned}
$$

$$\mathbf{E}(R(T)/T) \leq \left( \frac{D_\infty^2 G_\infty}{2\eta} + \frac{\eta G_\infty^2}{\sqrt{1-\beta}} \right) \sqrt{\frac{1+\sqrt{2d-1}}{2}\Delta_g + 1} \frac{d}{\sqrt{T}} + LD\sqrt{D^2 + \frac{d\alpha^2\Delta_w^2}{4}}.$$

$\square$

### A.4 Proof of Proposition 2

*Proof.* From Lemma 3,

$$\mathbf{E}(Q_g(\text{Clip}(\hat{\mathbf{g}}_t))) = \text{Clip}(\hat{\mathbf{g}}_t),$$

and

$$\mathbf{E}(\|Q_g(\text{Clip}(\hat{\mathbf{g}}_t))\|^2) \leq \mathbf{E}(\Delta_g\|\text{Clip}(\hat{\mathbf{g}}_t)\|_\infty\|\text{Clip}(\hat{\mathbf{g}}_t)\|_1 + \|\text{Clip}(\hat{\mathbf{g}}_t)\|^2).$$

As $[\hat{\mathbf{g}}_t]_i \sim \mathcal{N}(0,\sigma^2)$, its pdf is $f(x) = \frac{1}{\sqrt{2\pi\sigma^2}}e^{-\frac{x^2}{2\sigma^2}}$. Thus,

$$
\begin{aligned}
\mathbf{E}(\|\text{Clip}(\hat{\mathbf{g}}_t)\|_1) \leq \mathbf{E}(\|\hat{\mathbf{g}}_t\|_1) &= \sum_{i=1}^{d} \mathbf{E}(|[\hat{\mathbf{g}}_t]_i|) = d\int_{-\infty}^{+\infty} |x|f(x)\mathbf{d}x = 2d\int_{0}^{+\infty} xf(x)\mathbf{d}x \\
&= d\int_{0}^{+\infty} \frac{1}{\sqrt{2\pi\sigma^2}}e^{-\frac{x^2}{2\sigma^2}}\mathbf{d}x^2 = d\frac{-2\sigma^2}{\sqrt{2\pi\sigma^2}}e^{-\frac{u}{2\sigma^2}}\Big|_0^{+\infty} = (2/\pi)^{\frac{1}{2}}d\sigma.
\end{aligned}
$$

As $\mathbf{E}(\|\text{Clip}(\hat{\mathbf{g}}_t)\|^2) \leq \mathbf{E}(\|\hat{\mathbf{g}}_t\|^2)$, and that $\mathbf{E}(\|\hat{\mathbf{g}}_t\|^2) = d\sigma^2$, we have

$$
\begin{aligned}
\mathbf{E}(\|Q_g(\text{Clip}(\hat{\mathbf{g}}_t))\|^2) &\leq \mathbf{E}(\Delta_g\|\text{Clip}(\hat{\mathbf{g}}_t)\|_\infty\|\text{Clip}(\hat{\mathbf{g}}_t)\|_1 + \|\text{Clip}(\hat{\mathbf{g}}_t)\|^2) \\
&\leq \Delta_g c\sigma(2/\pi)^{\frac{1}{2}}d\sigma + \mathbf{E}(\|\text{Clip}(\hat{\mathbf{g}}_t)\|^2) \\
&\leq ((2/\pi)^{\frac{1}{2}}c\Delta_g + 1)\mathbf{E}(\|\hat{\mathbf{g}}_t\|^2).
\end{aligned}
$$

$\square$

### A.5 Proof of Proposition 3

*Proof.*

$$
\begin{aligned}
\mathbf{E}(\|\text{Clip}(\hat{\mathbf{g}}_t) - \hat{\mathbf{g}}_t\|^2) &= d\left( \int_{c\sigma}^{+\infty} (x-c\sigma)^2 f(x)\mathbf{d}x + \int_{-\infty}^{-c\sigma} (-x-c\sigma)^2 f(x)\mathbf{d}x \right) \\
&\leq 2d\int_{c\sigma}^{+\infty} (x-c\sigma)^2 f(x)\mathbf{d}x \\
&= \frac{2d}{\sqrt{2\pi\sigma^2}}\sigma^3\left(-ce^{-\frac{c^2}{2}} + \sqrt{\frac{\pi}{2}}(1+c^2)(1-\text{erf}(\frac{c}{\sqrt{2}}))\right) \\
&= (2/\pi)^{\frac{1}{2}}d\sigma^2 F(c).
\end{aligned}
$$

$\square$

### A.6 Proof of Theorem 3

*Proof.* When both weights and gradients are quantized, and gradient clipping is applied before gradient quantization, the update then becomes

$$\mathbf{w}_{t+1} = \mathbf{w}_t - \eta_t \text{Diag}(\sqrt{\check{\mathbf{v}}_t})^{-1}\check{\mathbf{g}}_t. \tag{24}$$

Similar to the proof of Theorem 1, and using that $\mathbf{E}(Q_g(\text{Clip}(\hat{\mathbf{g}}_t))) = \text{Clip}(\hat{\mathbf{g}}_t)$, we have

$$
\begin{aligned}
\mathbf{E}(R(T)) \leq &\sum_{i=1}^{d} \mathbf{E}\left(\frac{\sqrt{\check{v}_{1,i}}}{2\eta_1}(w_{1,i}-w_i^*)^2\right) + \sum_{i=1}^{d}\sum_{t=2}^{T} \mathbf{E}\left( (\frac{\sqrt{\check{v}_{t,i}}}{2\eta_t} - \frac{\sqrt{\check{v}_{t-1,i}}}{2\eta_{t-1}})(w_{t,i}-w_i^*)^2 \right) \\
&+ \sum_{t=1}^{T} \mathbf{E}(\langle \mathbf{w}_t - \mathbf{w}^*, \mathbf{g}_t - \check{\mathbf{g}}_t \rangle) + \sum_{i=1}^{d}\sum_{t=1}^{T} \frac{\eta_t}{2}\mathbf{E}\left( \frac{\check{g}_{t,i}^2}{\sqrt{(1-\beta)\check{g}_{t,i}^2}} \right)
\end{aligned}
$$

$$\leq \quad \frac{D_\infty^2}{2\eta}\sum_{i=1}^{d}\mathbf{E}(\sqrt{T\check{v}_{T,i}}) + \sum_{t=1}^{T}\mathbf{E}\langle\mathbf{w}_t - \mathbf{w}^*, \mathbf{g}_t - \hat{\mathbf{g}}_t\rangle + \sum_{t=1}^{T}\mathbf{E}\langle\mathbf{w}_t - \mathbf{w}^*, \hat{\mathbf{g}}_t - \text{Clip}(\hat{\mathbf{g}}_t)\rangle$$

$$+ \frac{\eta G_\infty}{\sqrt{1-\beta}}\sum_{i=1}^{d}\mathbf{E}(\|\check{\mathbf{g}}_{1:T,i}\|)$$

$$\leq \quad \frac{D_\infty^2}{2\eta}\sum_{i=1}^{d}\mathbf{E}(\sqrt{T\check{v}_{T,i}}) + \sum_{t=1}^{T}\mathbf{E}\langle\mathbf{H}_t^{'\frac{1}{2}}(\mathbf{w}_t - \mathbf{w}^*), \mathbf{H}_t^{'-\frac{1}{2}}(\mathbf{g}_t - \hat{\mathbf{g}}_t)\rangle$$

$$+ \sum_{t=1}^{T}\mathbf{E}\langle\mathbf{w}_t - \mathbf{w}^*, \hat{\mathbf{g}}_t - \text{Clip}(\hat{\mathbf{g}}_t)\rangle + \frac{\eta G_\infty}{\sqrt{1-\beta}}\sum_{i=1}^{d}\mathbf{E}(\|\check{\mathbf{g}}_{1:T,i}\|)$$

$$\leq \quad \frac{D_\infty^2\sqrt{dT}}{2\eta}\mathbf{E}(\|\sqrt{\check{\mathbf{v}}_T}\|) + \frac{\eta G_\infty}{\sqrt{1-\beta}}\sum_{i=1}^{d}\mathbf{E}(\|\check{\mathbf{g}}_{1:T,i}\|)$$

$$+ \sqrt{L}D\sum_{t=1}^{T}\mathbf{E}(\sqrt{\|\mathbf{w}_t - \hat{\mathbf{w}}_t\|_{\mathbf{H}_t'}^2}) + \sum_{t=1}^{T}\mathbf{E}(\sqrt{\|\mathbf{w}_t - \mathbf{w}^*\|^2}\sqrt{\|(\hat{\mathbf{g}}_t - \text{Clip}(\hat{\mathbf{g}}_t))\|^2})$$

$$\leq \quad \frac{D_\infty^2\sqrt{dT}}{2\eta}\sqrt{\sum_{t=1}^{T}(1-\beta)\beta^{T-t}\mathbf{E}(\|\check{\mathbf{g}}_t\|^2)} + \frac{\eta G_\infty\sqrt{d}}{\sqrt{1-\beta}}\sqrt{\sum_{t=1}^{T}\mathbf{E}(\|\check{\mathbf{g}}_t\|^2)}$$

$$+ \sqrt{L}D\sum_{t=1}^{T}\mathbf{E}(\sqrt{\|\mathbf{w}_t - \hat{\mathbf{w}}_t\|_{\mathbf{H}_t'}^2}) + D\sum_{t=1}^{T}\mathbf{E}(\sqrt{\|\hat{\mathbf{g}}_t - \text{Clip}(\hat{\mathbf{g}}_t)\|^2}).$$

(21) in the proof of Theorem 1 still holds. Similar to the proof of Theorem 2, using the domain bound assumption in Section 3.1 (i.e., $\|\mathbf{w}_m - \mathbf{w}_n\| \leq D$ and $\|\mathbf{w}_m - \mathbf{w}_n\|_\infty \leq D_\infty$ for any $\mathbf{w}_m, \mathbf{w}_n \in \mathcal{S}$), and Proposition 3, we have

$$\mathbf{E}(R(T)/T) \quad \leq \quad \left(\frac{D_\infty^2 G_\infty}{2\eta} + \frac{\eta G_\infty^2}{\sqrt{1-\beta}}\right)\sqrt{(2/\pi)^{\frac{1}{2}}c\Delta_g + 1}\frac{d}{\sqrt{T}}$$

$$+ LD\sqrt{D^2 + \frac{d\alpha^2\Delta_w^2}{4}} + \sqrt{d}D\sigma(2/\pi)^{\frac{1}{4}}\sqrt{F(c)}.$$

$\square$

