# OpenReview forum: "Analysis of Quantized Models"
_ICLR.cc/2019/Conference_

### Official Review · AnonReviewer3 · 2018-10-25
**The title is misleading but paper contains good material.**

**Rating:** 7
**Confidence:** 4

**Review:**

In a distributed learning system where a parameter server maintains a full resolution copy of the parameters, communication costs can be reduced by (a) discretizing the weights that the server broadcasts to the workers, and (b) discretizing the gradients that the workers return to the parameter server. Following existing literature, the authors propose to discretize the parameters in a manner that limits its impact on the loss function by means of a diagonal approximation of the Hessian. This also means that one can bound the difference between the gradient for the full precision parameter and the gradient for the discretized parameter.  In contrast, they discretize the gradients stochastically so that the discretized version is an unbiased estimator of the full precision stochastic gradient. Since the stochastic gradient is itself an unbiased estimator of the gradient, this means we are dealing with an estimator whose variance has increased in a manner we can bound as well. The theoretical analysis consists in pushing these two bounds through classical analyses of the stochastic gradient algorithm, in this case, a regret-based version in the style of Zinkevich or Duchi.  Although i did not check the minute details of the proof, the argument feels correct and familiar.  They also give an interesting result in favor of clipping gradients, worth developing.

Although the title promises an analysis that holds for deep networks, this analysis strictly applies only to convex models. The author argue that the predictions made by this analysis also apply to deep networks, and support this argument with extensive experiments (which certainly represent a fair amount of work).  This result is believable but should not be construed as an analysis. Nevertheless, both results (the theoretical result for convex model and the empirical result for deep networks) are interesting and worth sharing.

The main caveat comes from the style the parallel learning algorithm they are considering.  In the data-parallel case (which they consider), parameter servers approaches have been displaced by setups where all workers update their copy of the weights using the allReduced gradients.  One could also use discretized gradients to speedup the allReduce operation (this is less of a win because latencies dominate) but this would only result in an increased variance and a much simpler analysis.

Finally I am not completely up-to-date with this line of work and cannot evaluate the novelty with confidence. This was not known to me, which is only a piece of evidence.

-- bumping down my score because the misleading title was not addressed by the author response.
-- bumping it up again because the authors have reacted.

---

> ### Author Response · Authors · 2018-11-23
> **RE: AnonReviewer3's review**
>
> 1. "analysis strictly applies only to convex models"
> - Please see our reply to Q3(b) for Reviewer 2 above.
>
> 2. "In the data-parallel case (which they consider), parameter server approaches have been displaced by setups where all workers update their copy of the weights using the allReduced gradients. One could also use discretized gradients to speedup the allReduce operation (this is less of a win because latencies dominate) but this would only result in an increased variance and a much simpler analysis."
> - For the all-reduce communication model with N=2^k workers, we need k steps to gather the gradients. Assuming that the gradients are directly averaged at each all-reduce step, our theory is independent of how the gradients are aggregated, and can be directly applied. The only difference with the parameter server model is that the number of bits used for gradients increases by one at every aggregation step in the worst case. However, even considering this effect, gradient quantization still significantly reduces the network communication time, as shown in Figure 4.

---

> > ### Comment · AnonReviewer3 · 2018-11-23
> > **The title is still misleading!**
> >
> > Please do not overlook the part of my reviews that explains that the title is misleading!  Your reply amounts to saying that others also use convex analyzes to justify what they propose. But they do not generally use a title that suggests that they theoretically analyze deep networks. In short I do not believe that the paper should be published with such a misleading title.
> >
> >
> > Another limitation is the fact that communication can be designed to overlap computation, i.e. communicate the gradients for one layer while computing the gradients for the previous one.  For instance, in CNNs, there is a lot of computation for relatively few gradients (because the kernels are smaller than the images). In such a common setup, it is unclear that quantizing the gradients brings much benefit.

---

> > > ### Author Response · Authors · 2018-11-24
> > > **Reply to misleading title**
> > >
> > > Thanks very much for your review and your feedback. Your review on the misleading title is very important, and we have discussed among the authors and changed it to "Analysis of Quantized Models" in our first revision. I am very sorry that I forgot to update manuscript immediately after the response,  and forgot to mention this in our initial response, and caused the misunderstanding.
> > >
> > > We are also thinking about the "computation and communication overlap" problem and will respond to you as soon as possible. Thanks again for your effort in improving our paper.  If you have any further questions or suggestions, please do not hesitate to let us know. We are willing to incorporate further suggestions or requests to improve our paper.

---

> > > > ### Author Response · Authors · 2018-11-25
> > > > **Reply to the "computation and communication overlap" problem**
> > > >
> > > > Thanks for your insightful comment.
> > > >
> > > > Gradient quantization (Seide et al., 2014; Wen et al., 2017; Alistarh et al., 2017; Bernstein et al., 2018) and gradient sparsification (Aji & Heafield, 2017; Wangni et al., 2017) are most useful when the communication cost is larger than the computational cost. This is the case, for example, in
> > > > (1) networks such as AlexNet, VGG and LSTM (Alistarh et al., 2017), in which most of the parameters are in the fully-connected layers (e.g., ~95% parameters for AlexNet, ~90% parameters for VGG-16, and almost all parameters for LSTM); or
> > > > (2) when the communication bandwidth is limited, e.g., on mobile devices.
> > > >
> > > > Assume that one single worker requires T training iterations, and each iteration takes t_comp time. The total training time is t_comp*T. With N workers, each processing 1/N of training samples (weak scaling), the number of training iterations required is T/N. When communication overlaps with computation (as suggested by the reviewer), the communication time is further reduced when gradient quantization is used. In the case where communication completely overlaps with computation, and computation takes longer than communication, the total training time is t_comp*(T/N). Here, we ignore the communication time for bottom layer gradients. The speedup compared to the single worker case is thus N, which is linear with the number of workers.

---

### Official Review · AnonReviewer1 · 2018-11-02
**Pedagogical though incremental contribution**

**Rating:** 7
**Confidence:** 4

**Review:**

Summary
------

The authors proposes an analysis of the effect of simultaneously quantizing the weights and gradients in training a parametrized model in a fully-synchronized distributed environment, using RMSProp training updates.

The authors provide a theoretical analysis in term of regret bound, when the objective functions are smooth, convex and gradient-bounded wrt the parameter. They also assume that the parameters remains in a compact space. Their conclusions are as follow (thm 1, 2 and 3):

- weight quantization, which is deterministic and therefore introduces a bias in the objective functions, introduces a non-vanishing term in the average reget, that depens on the quantization error, where the vanishing term decreases in O(d /sqrt(T)).

- gradient quantization, which is performed in a stochastic, unbiased way (wrt to the full-precision gradient) do not introduce a further non-vanishing term, but augments the constant factor in the vanishing term.

- gradient clipping onto gradient quantization reduced this constant factor, at the cost of ntroducing a further non-vanishing term in the average regret.

An experimental setting is performed to assess how much the theoretical conclusions derived ina simpe setting apply to predictive functions parametrized with neural-network. The experiments are three folded:
- a first toy experiment with convex objective validates the theoretical findings
- a second experiment performed on CIFAR assess the performance on a grid of weight/gradient quantization with or without gradient clipping
- a third experiement, that is profiled (synthetically) assesses the performance of wieght/gradient quantization when training a model on imagenet.

In conclusion, the authors observe that quantizing weight/gradients systematically lead to a slight decrease in performance but provides promising improvement in term of training speed

Review
------

The paper is well written, documented and well-sectioned, with well written theoretical guarantees and thorough experiments, including one on a large dataset. The theoretical guarantees are relatively non-surprising and their proofs are indeed little involved. The authors are yet the first to analyse the effect of biased weight quantization on one hand, and of gradient clipping on the other hand.

The reviewer would have appreciated further comparison with existing analysis, in particular a comparison between stochastic weight quantization and loss-aware deterministic weight quantization. The bias introduced by the latter seems the culprit in the reduction of predictive performance. What if we applied non-biased weight quantization, with stochastic quantized gradient ?

The experiments as presented are a little underwhelming: first of all, there is no report of training time on ImageNet, and I believe that the profiling as been made in a communication model and not in a real setting. It would be great to see the best training time that you achieve by weight/gradient quantization (say on 4 bits).

Moreover, it appears that even with 4 bit quantization, the test accuracy of the trained model is significantly reduced. Why not increase the size to say 6 or 8 bits ?

On a related aspect, can the communication quantization be used jointly with a forward/backward quantized evalution ?

Overall, although this paper is relatively incremental and has underwhelming experiments, it is a thorough work that is worthy of being presented at ICLR 2019, in the reviewer's opinion.

Minor
-----

p 2: the notation w_i is overloaded

Eq 1: S_w^d should read (S_w)^d (cartesian product)

Thm 3: the notation R() is overloaded

Figure 1 is very hard to read: increase the font size

Figure 3 4 6: increase the legend size, ensure that the color used vary in lightness for printing

Table 1: use bold font to indicate the best performing FP/FP model, and your best performing model

Fig 7 c: training curve

---

> ### Author Response · Authors · 2018-11-23
> **RE: AnonReviewer1's review[1/2]**
>
> Thanks for your review and suggestions.
>
> 1.(a)"comparison between stochastic weight quantization and loss-aware deterministic weight quantization"
> - Comparison of stochastic weight quantization (Theorems 4-6 in Appendix D) and loss-aware deterministic weight quantization (Theorems 1-3):
>      (1) Convergence speed: Similar to loss-aware weight quantization,
>          (i) Stochastic weight quantization converges with a O(d/\sqrt{T}) rate to the error (but with a different scaling, see
>               (17) and (25));
>          (ii) Gradient quantization slows convergence (relative to using full-precision gradients) by a factor related to
>                gradient quantization resolution \Delta_g and d; and
>          (iii) Gradient clipping makes the speed degradation dimension-free.
>     (2) Error: Stochastic weight quantization also has an error term LD\sqrt{dD_{\infty}^2\Delta_w^2/4}, which is related
>           to the weight quantization resolution and dimension. Moreover, this term can potentially be larger than the one
>           (LD\sqrt{D^2+ d\alpha^2\Delta_w^2/4}) induced by loss-aware weight quantization, as D_{\infty} can be much
>           larger than \alpha and dD_{\infty}^2 be much larger than D^2.
> - We now add an experiment on stochastic weight quantization (W-SQ4) on CIFAR-10 with two workers. The setting is the same as the CIFAR-10 experiment in Table 1. The numbers are test set accuracies (%). Compared to LAQ4 in Table 1, stochastic weight quantization has worse accuracies than the full-precision baseline, even with 4-bit weights.
>
> -----------------------------------------
> G		                        W(SQ4)
> -----------------------------------------
> FP		                        83.29
> SQ2(no clipping)	        81.31
> SQ2(clip, c=3)		82.80
> SQ3(no clipping		82.82
> SQ3(clip, c=3)		82.99
> SQ4(no clipping)	        82.92
> SQ4(clip, c=3)		82.90
> ----------------------------------------
>
> (b) "The bias introduced by the latter seems the culprit in the reduction of predictive performance. What if we applied non-biased weight quantization, with stochastic quantized gradient?"
> - If non-biased stochastic weight quantization with stochastic quantized gradient is applied, the resultant gradient w.r.t. the stochastically quantized weight is unbiased only for the linear model (Zhang et al., 2017).
> - For nonlinear models, stochastic weight quantization makes the gradient biased, and there is an induced error. Detailed bounds for linear stochastic weight quantization with full-precision gradient, quantized gradient with/without clipping can be found in Appendix D.
>
> 2.(a) "there is no report of training time on ImageNet, profiling as been made in a communication model and not in a real setting."
> - Profiling is based on a performance model which is commonly used in the gradient compression literature (e.g., Wen et al, 2017, "Deep gradient compression: Reducing the communication bandwidth for distributed training", ICLR-2018)
>
> (b) "It would be great to see the best training time that you achieve by weight/gradient quantization (say on 4 bits)."
> - To measure the actual computation time, dedicated hardware for low-bit operations are needed. This will be investigated in the future.
>
> 3. "it appears that even with 4 bit quantization, the test accuracy of the trained model is significantly reduced. Why not increase the size to say 6 or 8 bits?"
> - As suggested by the reviewer, we now add the results for 6-bit weight quantization (LAQ6) on the ImageNet dataset. Compared with Table 3, using 6 bits has comparable or slightly better accuracy than 4-bit. In particular, LAQ6 using quantized clipped gradients has less than 1% absolute top-1 accuracy drop compared to using full-precision weights.
>
> -----------------------------------------------------------------------------------------------------------------------------
> Weight      Gradient                   N=2         		             N=4                               N=8
> -----------------------------------------------------------------------------------------------------------------------------
> 				                      top1/top5  acc (%)      top1/top5  acc (%)      top1/top5 acc (%)
> -----------------------------------------------------------------------------------------------------------------------------
>                   FP                               54.23/77.54		     54.31/77.46                  54.75/78.18
> LAQ6        SQ3(no clipping)      52.64/76.08 		     53.00/76.38                  53.08/73.34
>                   SQ3(clip, c=3)           54.21/77.32 		     54.53/77.85                  54.61/78.10
> -----------------------------------------------------------------------------------------------------------------------------

---

> > ### Author Response · Authors · 2018-11-23
> > **RE: AnonReviewer1's review[2/2]**
> >
> > 4. "can the communication quantization be used jointly with a forward/backward quantized evaluation?"
> > - Our communication quantization is already used jointly with a forward quantized evaluation (as the quantized weights are used for forward propagation). However, currently, we do not consider quantizing the gradients in the backward propagation.

---

### Official Review · AnonReviewer2 · 2018-11-05
**interesting problem setting and analysis, unclear conclusion from the analysis and experiments**

**Rating:** 6
**Confidence:** 4

**Review:**


Summary:

This paper studies the convergence properties of loss-aware weight quantization with different gradient precisions in the distributed environment, in which servers keeps the full-precision weights and workers keeps quantized weights. The authors provided convergence analysis for weight quantization with full-precision, quantized and quantized clipped gradients. Specifically, they find that: 1) the regret of loss-aware weight quantization with full-precision gradient converge to an error related to the weight quantization resolution and dimension d. 2) gradient quantization slows the convergence by a factor related to gradient quantization resolution and dimension d. 3) gradient clipping renders the speed degradation dimension-free.

Comments:

Pros:

- The paper is generally well written and organized. The notation is clean and consistent. Detailed proofs can be found in the appendix, the reader can appreciate the main results without getting lost in details.

- The paper provides theoretical analysis for the convergence properties of loss-aware weight quantization with full-precision gradients, quantized gradient and clipped quantized gradient, which extends existing analysis beyond full-precision gradients, which could be useful for distributed training with limited bandwidth.

Cons:

- It is unclear what problems the authors try to solve. The problem is about gradient compression, or how the gradient precision will affect the convergence for training quantized nets in the distributed environment, in which workers have limited computation power and the network bandwidth is limited. It is an interesting setting, however, the author does not make it clear the questions they are asking and how the theoretical results can guide the practical algorithm design.

- The authors mentioned that quantized gradient slows convergence (relative to using full-precision gradient) in contribution 2 while also claims that quantizing gradients can significantly speed up training of quantized weights in contribution 4, which is contradictory to each other.

- It is not clear what relaxation was made on the assumptions of f_t in section 3.1. The analysis are still based on three common assumptions: 1) f_t is convex 2) f_t is twice differentiable 3) f_t has bounded gradients. The assumptions and theoretical results may not hold for non-convex deep nets. E.g., the author does not valides the theorems results on d with neural networks but only with linear models in section 4.1.

- The author demonstrate training quantized nets in the distributed environment with quantized gradients, however, no comparison is made with other related works (e.g., Wen et al, 2017).

Questions:

- Theorem 1 is an analysis for training with quantized weights and full-precision gradients, which is essentially the same setting as BinaryConnect. Similar analysis has been done in Li et al, 2017. What is the difference or connection with their bound?

- It is not clear how gradienta are calculated w.r.t. quantized weights on worker, is straight through estimator (STE) used for backpropagation through Q_w?

- In section 3.3, why is \tilde{g}_t stochastically quantized gradient? How about statiscally quantized gradients?

- Why do the authors use linear model in section 4.1? Why are the solid lines in Figure 3 finished earlier than dashed lines? For neural networks, a common observation is that the larger the dimension d, the better the generalization performance. However, Figure 3 and Theorem 1 seem to be contradictory to this common belief. Would it possible to verify the theorem on deep nets of different dimension?

- Why does the number of worker affect the performance? I failed to see why the number of workers affect the performance of training if it is a synchronized distributed training with the same total batch size. After checking appendix C, I think it is better to discuss the influence of batch sizes rather than the number of workers.

- Why is zero weight decay used for CIFAR-10 experiment but non-zero weight decay for imagenet experiment? How was weight decay applied in Adam for quantized weights?

Minor issues:
- The notation of full-precision gradient w.r.t quantized weights in Figure 1 should be \hat{g}_t, however, g_t is used.

---

> ### Author Response · Authors · 2018-11-23
> **RE: AnonReviewer2's review[1/3]**
>
> Thanks for your review and suggestions.
>
> 1.(a) "unclear what problems the authors try to solve."
> - The problem is about how the gradient precision affects convergence of weight-quantized nets in a distributed environment.
>
> (b) "the author does not make it clear the questions they are asking"
> - The question we want to study is: What are the convergence properties of networks with quantized weights and quantized gradients?
>
> (c) "how the theoretical results can guide the practical algorithm design"
> - The theoretical results show that gradient clipping should be used in training weight-quantized models with quantized gradients. Specifically,
>     (1). directly quantizing gradients (Section 3.3) slows convergence (compared to the original full-precision baseline in
>           Section 3.2) by a factor related to gradient quantization resolution \Delta_g and dimension d. This is problematic
>           as  (i) deep networks typically have a large d; and (ii) distributed learning often uses a small number of bits for
>           the gradients, and thus a large \Delta_g.
>     (2). gradient clipping makes speed degradation negligible (Section 3.4).
>
>
> 2. "The authors mentioned that quantized gradient slows convergence (relative to using full-precision gradient) in contribution 2 while also claims that quantizing gradients can significantly speed up training of quantized weights in contribution 4, which is contradictory to each other."
> - In contribution 2, convergence speed is measured by how fast the regret is reduced w.r.t. the number of gradient evaluations. Quantized gradient loses information and so requires more gradient evaluations. This can be alleviated by gradient clipping.
> - In contribution 4, we mean the total training time (computation time plus communication time) in a distributed learning setting. Quantizing gradient reduces the communication cost, and thus speeds up training.
> - Combining with the above two, training weight-quantized networks with clipped quantized gradients is fast (as can be seen from Figure 4).
>
>
> 3.(a) "not clear what relaxation was made on the assumptions of f_t in section 3.1."
> - Existing analysis assumes square loss on linear model (f_t) and unbiased gradient (Zhang et al., 2017), stochastic weight quantization (Li et al., 2017; De Sa et al., 2018) or simple deterministic weight quantization using the sign (Li et al., 2017). These limitations are relaxed in this paper.
>
> (b) "The assumptions and theoretical results may not hold for non-convex deep nets"
> - As mentioned in Section 3.1, the convexity assumption does not hold for nonconvex deep nets.
> - However, this assumption facilitates analysis of deep learning models, and has also been used in various papers (Kingma & Ba, 2015; Reddi et al., 2018; Li et al., 2017; De Sa et al., 2018).
> - Moreover, as can be seen from Section 4, it helps to explain the empirical behavior.
>
> (c) "the author does not validate the theorems results on d with neural networks but only with linear models in section 4.1."
> - As mentioned in section 4.1, popular deep networks usually have hand-crafted architectures, and thus we used the linear model (which is also used in Zhang et al., 2017) in the submission.
> - As suggested by the reviewer, we added an experiment that varies the dimension of deep networks on CIFAR-10 dataset. The results can be checked at
>   https://www.dropbox.com/s/bcwwzu35fu496yv/iclr19_rebuttal.pdf?dl=0
> Similar to the linear model results, a larger d leads to larger convergence speed degradation.
>
> 4. "no comparison is made with other related works (e.g., Wen et al, 2017)"
> - Indeed, we have compared with (Wen et al, 2017) and (Alistarh et al., 2017). Note that in Table 1, SQ2 corresponds to Terngrad (proposed in Wen et al, 2017) with 2-bit stochastic quantization, and SQm corresponds to QSGD (proposed in Alistarh et al., 2017) with m-bit quantization.

---

> > ### Author Response · Authors · 2018-11-23
> > **RE: AnonReviewer2's review[2/3]**
> >
> > 5. "Theorem 1 is an analysis for training with quantized weights and full-precision gradients, which is essentially the same setting as BinaryConnect. Similar analysis has been done in Li et al, 2017. What is the difference or connection with their bound?"
> > - We differ from the setting in BinaryConnect as:
> >     (1) in BinaryConnect, there is no scaling parameter \alpha; whereas in loss-aware weight quantization, there is
> >           a \alpha learned from the data;
> >     (2) BinaryConnect considers only binarization, while we consider m-bit quantization which is more general.
> > - We differ from the bound in Li et al., 2017 as:
> >     (1) Li et al, 2017 studies simple SGD; whereas loss-aware quantization considers the proximal step with
> >           preconditioning (Section 2.2) like RMSprop or Adam.
> >     (2) If we do not derive the last step in (12) (proof for Theorem 1 in Appendix B.1), then the convergence speed
> >           for loss-aware weight quantization is controlled by the two terms \frac{D_{\infty}^2}{2 \eta}\sum_{i=1}^{d}
> >          \sqrt{T\hat{v}_{T,i}} and \frac{\eta G_{\infty} }{\sqrt{(1-\beta)}} \sum_{i=1}^{d} ||\hat\g_{1:T,i}||_2. When
> >          the data features are sparse, these two terms can be much smaller than the O(d\sqrt{T}) bound (Duchi et
> >          al., 2011; Kingma & Ba, 2015) in Theorem 3 (Note that D^2 and G^2 are upper bounded by dD_{\infty}^2 and
> >          d G_{\infty}^2) in Li et al, 2017. This indicates faster convergence of loss-aware weight quantization than
> >          BinaryConnect in Li et al, 2017.
> >     (3) BinaryConnect in Li et al, 2017 uses the simple sign function for binarization; while loss-aware weight
> >           quantization minimizes a weighted distance with the full-precision weight.
> > - Their error bounds are special cases of ours. Specifically, when (i) \alpha is fixed at \Delta; (ii) \H'_t=LI, where L is the Lipschitz constant in Assumption 2 and I is the identity matrix; and (iii) one does not consider that the updated full-precision weight is outside the representable range (equation (14) in the submission), then (S_w)^d=\{-1, +1\}^d, \Delta_w=1, and the error LD\sqrt{D^2+d \alpha^2 \Delta_w^2/4} in our Theorem 1 reduces to \sqrt{d}LD\Delta_w/4, which is tighter than \sqrt{d}LD\Delta in Theorem 3 in Li et al., 2017.
> >
> > 6. "not clear how gradients are calculated w.r.t. quantized weights on worker, is straight through estimator (STE) used for backpropagation through Q_w?"
> > - As quantized weights are used in forward propagation, we can directly obtain the gradients w.r.t. these quantized weights. Hence, no STE is needed.
> >
> > 7. "why is \tilde{g}_t stochastically quantized gradient? How about statically quantized gradients?
> > - Recent gradient quantization papers (Wen et al., 2017; Alistarh et al., 2017, Zhang et al., 2017) all require the quantized gradient to be unbiased and all use stochastically quantized gradients.  Deterministic gradient quantization makes the quantized gradient biased.
> >
> > 8.(a) "Why do the authors use linear model in section 4.1?"
> > - Please see our reply to Q3(c) above.
> >
> > (b) "Why are the solid lines in Figure 3 finished earlier than dashed lines?"
> > - We use early stopping, and training is terminated when the average training loss does not decrease for 5000 iterations.
> >
> > (c) "Would it possible to verify the theorem on deep nets of different dimension?"
> > - As suggested by the reviewer, we added this experiment on neural networks, please see our reply to Q3(c) above.
> >
> > 9.(a) "Why does the number of worker affect the performance?"
> > - There are two scaling schemes in distributed training with data parallelism: strong scaling and weak scaling ("A framework for performance modeling and prediction", ACM/IEEE conference on Supercomputing, 2002). Here, we consider weak scaling, which is more popular in deep network training.
> > - In weak scaling, the same data set size is used for each worker. The gradients are averaged over the N workers as g_t = \frac{1}{N} \sum_{n=1}^{N} g_t^{(n)} (Here, with a slight abuse of notation, g_t ^{(n)} can be the full-precision gradient or quantized gradient with/without clipping). If the gradients before averaging are independent random variables with zero mean, and ||\g_t^{(n)}||^2 is bounded by G^2, then |\g_t||^2 is bounded by G^2/N. From Theorems 1-3, the number of iterations for convergence is subsequently reduced by a factor of 1/N as compared to using a single worker.
> >
> > (b) "influence of batch sizes rather than the number of workers"
> > - When weak scaling is used, the total batch size (i.e., the total number of samples processed by all workers in each iteration) is proportional to the number of workers. The influence of the number of workers on performance is discussed in our reply to 9(a) and empirically in Section 4.2.4 and Section 4.3.

---

> > > ### Author Response · Authors · 2018-11-23
> > > **RE: AnonReviewer2's review[3/3]**
> > >
> > > 10.(a) "Why is zero weight decay used for CIFAR-10 experiment but non-zero weight decay for imagenet experiment?"
> > > - Weight quantization can be viewed as regularization (Courbariaux et al., 2015; Hou et al., 2017). For the small CIFAR-10, we do not need additional regularization such as weight decay (as also in Courbariaux et al., 2015; Hou et al., 2017; Hou et al, 2018).
> > > - For ImageNet, weight decay is also used in other weight-quantized networks (Li & Liu, 2017, Zhu et al., 2017, Leng et al., 2018).
> > >
> > > (b) "How was weight decay applied in Adam for quantized weights?"
> > > - The weight decay is used on the quantized weights during forward propagation.
> > >
> > > 11. "The notation of full-precision gradient w.r.t quantized weights in Figure 1 should be \hat{g}_t, however, g_t is used."
> > > - Thanks for pointing out this typo, and we have corrected it in the manuscript.

---

### Meta-Review · Area_Chair1 · 2018-12-12
**Good convergence analysis on convex model training with combined weight and gradient quantization, and empirical evidence for deep networks.**

**Confidence:** 3
**Recommendation:** Accept (Poster)

**Metareview:**

This paper provides the first convergence analysis for convex model distributed training with quantized weights and gradients. It is well written and organized. Extensive experiments are carried out beyond the assumption of convex models in the theoretical study.

Analysis with weight and gradient quantization has been separately studied, and this paper provides a combined analysis, which renders the contribution incremental.

As pointed out by R2 and R3, it is somewhat unclear under which problem setting, the proposed quantized training would help improve the convergence. The authors provide clarification in the feedback. It is important to include those, together with other explanations in the feedback, in the future revision.

Another limitation pointed out by R3 is that the theoretical analysis applies to convex models only. Nevertheless, it is nice to show in experiments that deep networks training is benefitted from the gradient quantization empirically.